# A Review of Oil–Solid Separation and Oil–Water Separation in Unconventional Heavy Oil Production Process

**DOI:** 10.3390/ijms24010074

**Published:** 2022-12-21

**Authors:** Xiao Xia, Jun Ma, Shuo Geng, Fei Liu, Mengqin Yao

**Affiliations:** 1Department of Chemical Engineering, School of Chemistry and Chemical Engineering, Guizhou University, Guiyang 550025, China; 2Guizhou Key Laboratory for Green Chemical and Clean Energy Technology, Guiyang 550025, China

**Keywords:** oil–solid separation, heavy oil ores, production process

## Abstract

Unconventional heavy oil ores (UHO) have been considered an important part of petroleum resources and an alternative source of chemicals and energy supply. Due to the participation of water and extractants, oil–solid separation (OSS) and oil–water separation (OWS) processes are inevitable in the industrial separation processes of UHO. Therefore, this critical review systematically reviews the basic theories of OSS and OWS, including solid wettability, contact angle, oil–solid interactions, structural characteristics of natural surfactants and interface characteristics of interfacially active asphaltene film. With the basic theories in mind, the corresponding OSS and OWS mechanisms are discussed. Finally, the present challenges and future research considerations are touched on to provide insights and theoretical fundamentals for OSS and OWS. Additionally, this critical review might even be useful for the provision of a framework of research prospects to guide future research directions in laboratories and industries that focus on the OSS and OWS processes in this important heavy oil production field.

## 1. Introduction

Energy and environmental issues have been paid constant attention. As one of the three major fossil energy sources and important chemical raw materials, petroleum resources play a major role in the development of human society and the economy [1]. However, with the large-scale exploitation and consumption of crude oil, the output of conventional oil resources is gradually decreasing [2]. This has become an important factor affecting the stable development of the world economy and society. With the decrease of conventional petroleum resources and the rise of production costs, the development and utilization of unconventional petroleum resources has attracted extensive attention [3]. Among these, the exploitation and utilization of unconventional heavy oil ores have also attracted the widespread concern of scientific researchers [4,5]. The distribution and reserves of unconventional heavy oil ores (UHO) are shown in Figure 1 [6]. Unconventional oil deposits, mostly located in North America, Eastern Europe and Latin America, are enormous with proven reserves of more than six trillion barrels of recoverable oil by current technologies, accounting for about 70% of total world oil reserves [7,8]. In China, unconventional oil reserves are relatively large and are mainly distributed in Inner Mongolia, Xinjiang, Shanxi, Heilongjiang, Liaoning provinces, etc. [9]. The proven reserves of oil shale resources are 311.7 × 10^8^ tons and oil sands reserves are 59.7 × 10^8^ tons [10]. Therefore, as an important part of fossil energy, UHO possesses great mining and utilization value. In addition, the rational development and utilization of UHO is of great significance in promoting the optimization of the global energy structure.

At present, some main separation methods have been used to separate heavy oil from UHO, mainly including pyrolysis [11], hot water-based extraction (HWBE) [12], solvent extraction (SE) [13,14] and reactive extraction (RE) [15]. The pyrolysis method works based on the energy-intensive reaction and high oil content in unconventional oil ores. The pyrolysis of UHO is a process of cracking heavy oil to produce volatiles through the fracture of chemical bonds at high temperature. The pyrolysis process produces gaseous and liquid products, for which the low boiling point components, known as dry gas in the industry, are difficult to condense leaving the high boiling point components to be condensed into liquid products, namely pyrolytic oil. This process is shown in Figure 2 [16]. HWBE mainly uses the penetration of surfactant to reduce the adsorption force of heavy oil on the sand surface before it undergoes floated oil treatment in a hot water environment [17]. Due to the density difference of each component, the separation of heavy oil and sands is realized. The separation processes are shown in Figure 3 [18], and mainly include two stages: heavy oil dissociation and air flotation. SE is based on the principle of ‘‘similar phase solubility” of organic solvents to dissolve the heavy oil components in UHO [19]. The process of separating UHO by SE is shown in Figure 4. RE is based on the reaction between UHO and acid solution to separate heavy oil [20]. The processes of separating heavy oil from UHO by the above methods include two important processes: oil–solid separation and oil–water separation. Clarifying the technical bottlenecks of these two processes will offer significant implications for the exploitation and recovery of UHO.

As shown in Figure 5. The oil–solid separation process of UHO is the separation of heavy oil and mineral solid particles. The key technical bottleneck of this process is how to quickly and effectively peel the heavy oil from the surface of mineral solid particles. The oil–water separation process of UHO is the separation of heavy oil and water. In the process of separating UHO by HWBE and RE, the participation of water is involved in different degrees and natural surfactants (asphaltenes, resins, naphthenic acid etc.), amphiphilic (hydrophilic and lipophilic) mineral solid particles in heavy oil. The stable heavy oil–water emulsion phase is formed [21]. The heavy oil–water emulsions will bring a series of problems to the subsequent oil separation and processing processes, such as an increase in processing cost, corrosion of pipelines and equipment, reduction in oil quality, pollution of the environment etc. [22]. Therefore, the question of how to efficiently break heavy oil–water emulsions is not only a universal technical bottleneck in the oil–water separation process of UHO, but also possesses important practical application value in heavy oil exploitation and processing. Next, some methods to solve the technical bottleneck in the process of oil–solid separation and oil–water separation are introduced. 

At present, in order to solve the bottleneck problem of low oil–solid separation efficiency in the process of UHO separation, the additive assisted solvent extraction is often used to improve the oil–solid separation efficiency. For example, Xingang Li et al. [23] used an ionic liquid (IL) with low viscosity, 1-ethyl-3-methyl imidazolium tetrafluoroborate ((Emim) (BF_4_)), to enhance bitumen recovery from oil sands by solvent extraction using a composite solvent of *n*-heptane and acetone. Their results demonstrate that (Emim) (BF_4_) increased the bitumen recovery by up to 95% at room temperature, with much less fine clay in the recovered bitumen than those using solvent extraction without IL, and with no organic residue observed in the spent sands. Hong Sui et al. [24] synthesized a novel form of process aid, switchable-hydrophilicity tertiary amines (SHTAs, such as triethylamine, N,N-dimethylcyclohexylamine, N,N-dimethylbenzylamine) in their hydrophilic forms, and applied them to the task of enhancing heavy hydrocarbon recovery from oil sands ores during solvent extraction. It has been observed that the secondary extraction rate of asphalt is 98%, which is 5% higher than that of pure solvent extraction. At the same time, the solid entrainment of asphalt oil and the solvent residue of extracted residual sand are greatly reduced. Yi Lu et al. [25] used CO_2_-responsive surfactants to enhance the overall heavy oil recovery by switching their interfacial activity to the desired state in each stage. In addition, physical demulsification, biological demulsification and chemical demulsification methods are often used for the separation of oil–water emulsions [26,27]. This is because chemical demulsification, as an economic and efficient demulsification method, has the advantages of high demulsification efficiency, low cost and wide application range [28]. Therefore, the chemical demulsification method is often used to separate heavy oil and water [29,30]. This method mainly added chemical demulsifier with strong interfacial activity into emulsions. Using the interaction between chemical demulsifier and emulsifier on an oil–water interface can change the physicochemical properties of the oil-water interface [31], further destroy the interfacial films, accelerate the aggregation, coalescence and sedimentation processes of emulsion droplets, and finally achieve the purpose of demulsification [32]. For example, Jun Ma et al. [33] successfully synthesized a novel aliphatic alcohol nonionic polyether (MJTJU-2) by esterification and polymerization. It was found that this new polyether is efficient in rapidly demulsifying the most stable heavy oil–water emulsion (interfacially active asphaltene (IAA)-stabilized emulsions) with almost complete dehydration (>97%) of the emulsions achievable in less than 15 min at 60 °C using 400 ppm of MJTJU-2. To sum up, the separation mechanisms of oil–solid separation and oil–water separation in unconventional heavy oil ore production processes are still unclear, and comprise a key scientific problem leading to the low efficiency of oil–solid separation and oil–water separation.

In this review article, as shown in Figure 6. An attempt is made to cover a wide range of research studies of oil–solid separation (OSS) and oil–water separation (OWS) in UHO production processes. First, an overview of the fundamental theories of OSS and OWS are presented. Following that, the research statuses of OSS and OWS mechanisms are reviewed. Eventually, we also share our views on the challenges and perspectives for the research of OSS and OWS in UHO field. This review aims to give comprehensive summaries of OSS and OWS in the UHO production process. To be more focused in its content, this article will not cover the development of organic extractants and chemical demulsifiers, for which interested readers should refer to other recent reviews [34,35,36,37,38].

## 2. Fundamental Theories of OSS and OWS 

Unconventional heavy oil normally coexists tightly with solids such as rocks, sands and clay minerals. Regardless of the separation methods, an obvious requirement for effective recovery of unconventional heavy oil from the corresponding UHO is the successful separation of unconventional heavy oil from solids, which is determined primarily by the oil–solid separation and oil–water separation processes. Therefore, an understanding the basic theories of OSS and OWS is the premise for the efficient recovery of heavy oil from UHO. In this section, we will start with a detailed summary of the basic theory of oil–solid separation, including solid wettability [39], contact angle and oil–solid interactions [40,41]. Then, the basic theory of oil–water separation will be reviewed, including the characteristics of oil–water interfacial films and oil–water interactions. For narrative convenience, the emulsions described below refer to heavy oil–water emulsions.

### 2.1. Basic Theories of Oil–Solid Separation

A large amount of research has shown that, in the oil–solid separation process of UHO, both the wettability of the solid surface and the interaction between the heavy oil and the solid play a major role in the oil–solid separation efficiency [42,43,44,45,46]. Therefore, in this section, for the basic theory of oil–solid separation we mainly review the solid wettability, contact angle and oil–solid interaction.

#### 2.1.1. Solid Wettability and Contact Angle

Wetting is a common phenomenon in nature. It is an interface effect caused by the extrusion of liquid and solid surfaces on the solid surface. A surface that is easily wetted is known as a hydrophilic (water) surface; on the contrary, one that is not easily wetted is known as a hydrophobic (water) surface. Studying the wettability of a solid by a liquid is essential in understanding the science of oil–solid separation. The wetting phenomenon in UHO refers to three phase systems, in which one phase must be solid, the other phase must be liquid, and the third phase is gas or liquid. In such a three-phase system, the contact angle is usually used to measure the wettability of the solid surface (hydrophilic or hydrophobic degree of solid surface). When bubbles adhere to the solid surface (or water droplets adhere to the solid surface), it is generally considered that the contact is the three-phase contact, and this contact line is called “three-phase wetting perimeter”. During the contact process, the wetted perimeter can move, or become larger or smaller. When the change is stopped, this indicates that the free energy (expressed in interfacial tension) of the three-phase interface has reached equilibrium. Under this condition, the included angle between the tangent line of the liquid–gas interface and the tangent line of the solid–liquid interface at any point on the wetting perimeter is called the equilibrium contact angle, referred to as the contact angle for short. The contact angle shown in Figure 7 for solid–water–air (a) and solid–oil–water systems is often used to characterize the wetting characteristic of solids in an oil–solid separation process. The relation between the contact angle in an equilibrium configuration (i.e., static or equilibrium contact angle) and the three interfacial tensions, in the simplest form, is described by the well-known Young’s equation [47], given in Equation (1): (a) for a solid–water–air system and (b) for a solid–oil–water system.
γ_s/a_ − γ_s/w_ = γ_w/a_·cosθ(1a)
γ_s/o_ − γ_s/w_ = γ_w/o_·cosθ(1b)

Here γ and θ represent the surface (or interfacial) tension and contact angle, respectively, while subscripts s, a, w and o denote solid, gas, water and oil phases, respectively.

It is noteworthy that the contact angle is normally measured from the aqueous phase if water is involved [48,49]. Solids in UHO with the water contact angle close to zero are easily wetted by water and classified as hydrophilic (water-wet). On the contrary, solids are considered hydrophobic (or oil-wet) if the water contact angle is larger than 90°. The solids of water contact angle around 90° are known to be bi-wettable, they are easily separated at the oil–water interface. 

#### 2.1.2. Oil–Solid Interactions

The interactions between oil molecules and solid surfaces are considered an important factor in control the separation of unconventional heavy oil from UHO. Experimentally, zeta potential and surface force measurements are often used to characterize the interactions between the oil molecules and solids [50]. Theoretically, the well-known Derjaguin–Landau–Verwey–Overbeek) (DLVO () and extended DLVO theories are applied to analyze interactions between the oil molecules and solids [51,52]. Silica sands, carbonate rocks and clay minerals are the three main classes of mineral solids in UHO. Up until now, the interactions between silica sands and heavy oil have been studied extensively [53,54,55]. In general, the interactions between silica sands and heavy oil are highly dependent on the water chemistry (e.g., pH, divalent cation concentration, and ionic strength) and temperature. Dai and Chung have observed, through dissociation experiment, that the adhesion of silica and heavy oil in low pH aqueous solutions is stronger [56,57]. The results of atomic force microscopy (AFM) surface force measurements also support this observation. Increasing the concentration of ions, such as Na^+^ and Ca^2+^, reduces repulsive forces. Increasing the concentration of ions can even reverse the repulsive forces to become attractive between bitumen and silica in alkaline solution, as a result of compressing the electric double layers around silica and bitumen surfaces or through specific adsorption on these surfaces [58]. In particular, when Ca^2+^ is present in the solution, strong agglomeration between heavy oil and silica can be observed. This phenomenon can be attributed to the fact that Ca^2+^ can be specifically adsorbed on the silica surface and become a bridge between silica particles and heavy oil in aqueous solution [59]. Due to the weathering of UHO, the surface hydrophobicity of mineral solids is enhanced, and the adhesion between solids and heavy oil is stronger, resulting in the reduction of heavy oil recovery rate [60,61]. The interaction between heavy oil and clays is similar to that of silica, but the adhesion force is stronger than that of silica [62,63]. Liu et al. [58] found that the adhesion forces of heavy-oil silica and heavy-oil clays were 0.3 mN/m and 3–9 mN/m respectively in an aqueous solution of 1 mm KCl and a pH of 8. In clay minerals, the adhesion forces between montmorillonite and heavy oil are very strong. 

In addition, the composition of heavy oil also directly affects the oil–solid interaction. Kumar et al. [64] found that polar components in heavy oil possess a stronger affinity for solid surfaces, making them more hydrophobic. For example, the asphaltene component in heavy oil—as an important type of interfacial active substance, the interaction between asphaltene and solid has also been widely considered. Abraham et al. [65] found that the repulsion between asphaltene and silica increases with an increase in pH of an aqueous solution, This indicates the presence of pH-dependent ionizable/polarized groups on the asphaltene surface. Compared with hydrocarbon oil components, there is a strong polar interaction between ionizable/polarized groups in asphaltene and silicon hydroxyl groups on the surface of silica [66,67], which enhances the affinity between asphaltene and solid surface. Given the above, we can see that the interaction between heavy oil components and mineral solids plays a key role in the separation system of UHO. Their interaction is directly affected by mineral type, heavy oil composition and water chemistry. 

### 2.2. Basic Theories of Oil–Water Separation

During the process of unconventional heavy oil, heavy oil–water emulsions are inevitably generated. The formation of the oil–water emulsions is due to following three conditions [68,69]. (1) There is an immiscible oil phase (heavy oil) and an immiscible water phase in the system; (2) The oil–water system contains one or more indigenous surface-active components of heavy oil (e.g., asphaltenes, resins and natural surfactants), mineral particles and externally added surfactants. (3) There is the requisite strength of mechanical shear force to make the immiscible oil phase and water phase fully mix and form emulsions. As shown in Figure 8, The emulsions can form in different types: water-in-oil (W/O), oil-in-water (O/W) and multiple emulsions, such as (oil-in-water)-in-oil (O/W/O) or (water-in-oil)-in-water (W/O/W) emulsions [70]. Theoretically, the emulsions are thermodynamically unstable, as a result of the excess energy of the interfaces. However, the heavy oil–water emulsions are kinetically stable for a long period of time, from several hours to several months or even longer. In addition, these emulsions will also significantly reduce the yield and quality of heavy oil [68]. 

To summarize, the study of the demulsification of heavy oil–water emulsions is not only a universal scientific and technical problem in the field of heavy oil, but also possesses important practical application value in the process of UHO exploitation and processing. However, clarifying the stability mechanism of heavy oil–water emulsions is the foundation of the efficient separation of heavy-oil water. In addition, a large number of studies have shown that the oil–water interfacial films formed by the natural surface-active components (asphaltenes, resins, naphthenic acid, amphiphilic solid particles) at the oil–water interface plays a major role in stabilizing heavy oil–water emulsions [71,72,73,74]. Therefore, in the following sections, the structural characteristics of natural surface-active components and the interface properties of oil–water interfacial films will be reviewed and discussed.

#### 2.2.1. Structural Characteristics of Natural Surfactants

Structural characteristics of asphaltenes: Asphaltene is a complex molecule and is the heaviest and the most aromatic fraction of the UHO. From an operational point of view, it is soluble in aromatic solvents such as benzene and toluene and insoluble in light alkanes such as n-pentane and n-heptane [75]. Elemental compositions of different asphaltenes have been reported in the related literature [76,77,78]. It has been revealed that typical asphaltenes are composed of carbon (C), hydrogen (H), oxygen (O), nitrogen (N), sulfur(S) as well as small amounts of iron (Fe), vanadium (V), and nickel (Ni) in ppm levels [79]. The latest research shows that the carbon and hydrogen content of heavy-oil asphaltene ranges from 81.32% to 83.40% and 6.80% to 8.80%, respectively. It can be concluded that the range of hydrogen/carbon ratio in heavy oil asphaltenes is 0.97 to 1.28. In comparison, heteroatoms including oxygen and sulfur have various concentrations including oxygen 0.98–2.85% and sulfur 5.20–8.15%. However, the nitrogen content shows less variation, which is in the range of 0.92–1.26% [80].

The molecular structure of asphaltene has been the subject of many research studies [81,82,83,84,85]. In early works, two dominant structure models were accepted to describe the molecular structure of the asphaltene: namely, the “island” and the “archipelago” models [86,87]. The “island” model is one in which there is only one fused aromatic ring system in the structural unit of the asphaltene molecule, with the fused aromatic ring as the center, multiple cycloalkyl rings are combined with it, and alkyl side chains with or without heteroatoms are connected with it. Its average molecular weight is about 750 g/mol. This “island” model possesses seven fused aromatic rings, and heteroatoms and metal elements in the form of porphyrins also exist in the fused aromatic ring system [81,88,89]. However, there are several similar “island” model structures in the structural units of “archipelagic” asphaltene molecules, and these structural units are connected by alkyl chains. The number of fused aromatic rings in the “archipelagic” model structure is far less than seven [90,91]. The “Yen–Mullins” model is considered one of the most acceptable molecular structures of asphaltenes (Figure 9) [92]. At present, the most likely molecular structure model of asphaltene to have been proposed by researchers is shown in Figure 10. The difference between this asphaltene molecular structure model and the above model is that the influence of metal atoms and electric charges on molecular structure characteristics of asphaltene is considered [93]. 

Structural characteristics of resins: Unlike asphaltenes, resins are components that are soluble in light alkanes. The structures of resins are similar to those of asphaltenes but with smaller molecular weight, higher H/C ratios and less heteroatoms in their structure [94,95]. The resin fraction is known to consist of C, H, N, O and S, the carbon and hydrogen mass content in molecules of heavy oil resins varies in a narrow range: 85 ± 3% C and 10.5 ± 1% H [96,97]. One of its important characteristics is its H/C atomic ratio, which varies within the range of 1.2–1.7 for resins of various oils, which is on average higher than that for asphaltenes, where H/C is from 0.97 to 1.28. Nitrogen and sulfur mass content are mainly 0.2–0.5% and 0.4–5.1% in resins. Resins are characterized by a higher oxygen content (1–10%) [98]. 

Instrumental analysis data have significantly extended the understanding of the structure of resin molecules [99]. Despite the lack of well-defined molecular structures, resins are mainly polar, polynuclear molecules composed of aromatic rings, aliphatic side chains and a few heteroatoms. A typical molecular structure of resins is shown in Figure 11 [100]. It has been claimed that resins contain a variety of water-loving functional groups, such as thiophene, benzothiophene, dibenzothiophene, pyrrole N-H, ester, carbonyl, hydroxyl, carboxyl and sulfonic acids [101,102]. 

Structural characteristics of naphthenic acid and amphiphilic solid particles: Naphthenic acids are weak-acid types of natural surfactants often contained in UHO [103]. They belong to a class of small molecular substances with a certain interfacial activity (due to their amphiphilic nature) [104]. Generally speaking, naphthenic acid (NA) refers collectively to a family of cycloaliphatic carboxylic acids present in heavy oil. These have an empirical formula of C_n_H_2n+z_O_2_, where n is the number of carbon and z is zero or even a negative integer representing the hydrogen deficiency (unsaturated degree) of an NA molecule [105]. The definition has recently been expanded to the naphthenic acid fraction component (NAFC), which includes unsaturated and aromatic NA derivatives, increased oxygen content, and compounds containing nitrogen and/or sulfur [106,107]. Typical structures of NAs are shown in Figure 12 [108]. The value of z can vary from 0 to −2, −4, −6, −8, −10 and −12. However, the latest research shows that the ratio of even to odd (E/O) carbon number NAs in the different heavy oils studied is close to 1.0 for z −2 to z −24 NAs. NA series z 0 to z −12 in bitumen account for about 90% of the total determined NAs, and z −14 to z −24 NAs make up the remaining ∼10% [109]. 

Generally speaking, amphiphilic (hydrophilicity and lipophilicity) solid particles in UHO tend to include carbonate and silicate. In carbonate materials, dolomite and calcite exhibit similar structures, wherein layers of carbonate (CO_3_^2−^) groups separate layers of different cations (Ca^2+^ in calcite, or Ca^2+^ and Mg^2+^ in dolomite) (see Figure 13) [110]. The basic structural unit of feldspar is tetrahedron, which is composed of four oxygen atoms surrounding one silicon atom or aluminum atom. Each tetrahedron shares an oxygen atom with another tetrahedron, forming a three-dimensional skeleton [111]. Magnesite is mainly composed of MgCO_3_, and has a rhombohedral system structure [112]. The phyllosilicates (sheet silicates) are an important class of silicate minerals, mainly including kaolinite, illite, montmorillonite, etc. [113,114,115]. These phyllosilicates consist of tetrahedral sheets (T: silicon–oxygen tetrahedron network) and octahedral sheets (O: six-fold coordination of aluminum with oxygen from the tetrahedral sheet and hydroxyl groups, while individual octahedral is linked laterally by sharing edges). Different stacking of T and O sheets leads to different types of clay minerals, such as trilayer (TOT) illite, montmorillonite, and bilayer (TO) kaolinite (Figure 14I) [116]. In addition, the interface adsorption structure characteristics of kaolinite, illite, montmorillonite are also affected by hydrogen bonds and charges (Figure 14II) [116].

#### 2.2.2. Interface Characteristics of Asphaltene Interfacial Films

At present, extensive research has shown that the oil–water interfacial films formed by the asphaltenes in heavy oil at the heavy oil–water interface play a major role in the stabilizing of heavy oil–water emulsions [117,118,119,120,121]. In addition, the other surface-active components (resins, naphthenic acid and amphiphilic solid particles) have an important influence on the interfacial characteristics of asphaltene interfacial films [122,123,124,125]. Therefore, based on the understanding of the structural characteristics of natural surface-active components in Section 2.2.1, the interface characteristics of asphaltene interfacial films are reviewed and discussed in detail in this section. 

Asphaltenes are surface active, both on solid surfaces and at liquid–liquid, especially in heavy oil–water interface, where they adsorb to stabilize heavy oil–water emulsions [126]. They self-assemble and entangle, forming discoidal nanoaggregates in solution and strong, stable films at oil–water interfaces [127,128,129,130,131]. These interfacial film aggregates of asphaltene are mainly formed through the intermolecular interactions of asphaltene [132]. Asphaltene–asphaltene intermolecular interactions principally comprise dipole–dipole, Coulombic, and van der Waals dispersion interactions, hydrogen bonding, and π–π stacking. In addition to these major contributors, steric and inductive interactions and intermolecular charge transfers are less common but still relevant [117,133,134,135,136]. The interfacial characteristics of asphaltene interfacial films are affected by the following factors. 

(1) Molecular weight of asphaltene [137]: Asphaltene belongs to a group known as autopolymers. The methods and solvents used to separate UHO will affect the molecular weight of asphaltene, resulting in different interface properties of asphaltene interfacial films. For asphaltenes with large molecular weight, their interfacial films usually exist in the form of aggregates, and their solubility is poor. However, due to the good solubility of asphaltenes with low molecular weight, the asphaltene interfacial films will be divided into several small aggregates and then completely dissolved. It can be seen that the asphaltenes with low molecular weight form their interfacial films in the form of small aggregates, while the asphaltenes with high molecular weight form their interfacial films in the form of large aggregates or clusters [138,139,140]. 

(2) pH value of the system: The pH value of a system has an effect on the interaction between asphaltene and asphaltene molecules [141]. For example, Clara et al. [142] used a Langmuir trough to evaluate asphaltene interfacial films (C5I) under compression at a constant rate by surface pressure-area isotherms. They found that the asphaltene monolayers displayed an extensive region containing liquid-expanded (LE) and liquid-condensed (LC) phases and a well-identified transition region between these phases. The compressibility of LE films (0.01–0.02 m/mN) was approximately five-fold lower than the compressibility of LC films (0.05–0.07 m/mN). The mixed film compressibility was between the LE and LC film compressibilities. At 10 °C, more compressed films were obtained at pH 4. At 30 °C, more compressed films were found at pH 8. Surface pressure (π) was evaluated as a function of the subphase pH and temperature. At 20 °C, the lowest π was between the pH 4 and 8 isotherms. The highest surface pressure was 52 mN/m at 10 °C and pH 8, which denotes higher repulsive forces acting in the film. This is due to the existence of an electron-rich group of conjugated fused aromatic rings in the molecular structure of asphaltene, and heteroatoms (N, O and S) which also contain electron-rich groups with lone pair electrons [143,144]. Therefore, at low pH, H^+^ can neutralize the negative electricity in an asphaltene molecule, resulting in the weakening or even disappearance of intermolecular electrostatic repulsion, and the enhancement of intramolecular and intermolecular association, which makes asphaltene easier to form aggregates, increase molecular volume and reduce rigidity. In addition, when the pH value is low, the carboxylic acid groups in asphaltene molecules are not easy to be ionized. At high pH value, the above effects are just the opposite, causing asphaltene to be more negatively charged, increasing the intermolecular repulsion, and their films tend to be more stretched and rigid [145,146].

(3) Temperature: The influence of temperature on the properties of asphaltene interfacial films is complex. Temperature has a certain influence on the interaction between asphaltene and asphaltene molecules (Van Der Waals interaction, hydrogen bond, etc.) on the oil–water interface [147]. With increasing temperature, the interaction between asphaltene molecules is destroyed, which reduces the size of asphaltene films and increases the solubility. On the contrary, the size of asphaltene film increases with the decrease of temperature [147,148].

The rheological properties are also important interface characteristics of asphaltene interfacial films at the oil–water interface. Asphaltenes adsorb to oil–water interfaces and form continuously rigid interfacial films on the order of 2–10 nm thick that stabilize heavy oil–water emulsions and resist deformation due to their viscoelastic properties by acting as physical barriers that can sterically inhibit droplet coalescence (see Figure 15) [149,150,151,152,153,154,155]. It is generally accepted that asphaltene adsorption leads to the formation of elastic skins around water droplets [156,157,158]. The viscoelastic properties of asphaltene-stabilized interfacial films depend on oil composition, physicochemical properties of asphaltenes, acidity, surface coverage (encoded in surface aging), time scale/frequency susceptibility to deformation, the dynamics of the fluids that surround the interface, and the interactions between adsorbed molecules [159,160,161,162,163]. As both surface elasticity and viscosity take place upon interfacial deformation, the mechanism driving molecular motion, configuration, orientation, and interactions influence the mechanical and rheological properties of the asphaltene stabilized interfacial films [164,165]. For example, Liyuan Feng et al. [166] investigated the impact of sodium citrate (Na_3_Cit) on the oil–water interface containing asphaltene. A rheometer equipped with a double wall ring (DWR) system was utilized to determine the shear rheological response of asphaltene film as a function of the subphase solution salinity. These results collectively suggest that Na_3_Cit–asphaltene interactions result in a looser and more elastic asphaltene interfacial network with a slow formation and reduces the adhesion between the two interfaces, all of which are most likely the consequence of increased electrostatic repulsion between asphaltene interfacial nanoaggregates. At present, numerous investigations have focused on understanding the viscoelastic behavior of asphaltene interfacial films using interfacial rheological measurements (e.g., shear, dilatational, and mixed-flow) and found that interfacial films’ properties vary with bulk concentration, aging time, solvent quality, and asphaltene origin (see Figure 16A) [88,150,167,168,169,170]. For example, as seen in Figure 16B, both the shear and dilatational elastic storage moduli increase with aging time, which is believed to be caused by interfacial rearrangement, crowding, and stronger intermolecular associations of asphaltenes in the interfacial films. The microstructure of asphaltene films influences interfacial shear thinning, fracture, and yielding behavior, where elastic films, multilayer aggregates, surface wrinkles, and dispersed aggregates are shown in Figure 16C.

## 3. Separation Mechanism

During oil production from unconventional oil ores, oil solid separation and oil–water separation are two inevitable processes. Understanding the separation mechanism of these two processes is of great significance for the efficient recovery of heavy oil. Therefore, in this section, the research status of oil–solid separation and oil–water separation mechanisms is systematically reviewed.

### 3.1. Oil–Solid Separation Mechanism

At present, in the separation process of UHO, the research on the mechanism of oil–solid separation has been reported. For example, as shown in Figure 17, Zisheng Zhang et al. [43] synthesized a novel amino acid ionic liquid-based deep eutectic solvent (TrpBF_4_/U) by using L-tryptophan, fluoroboric acid and urea. It is found that the TrpBF_4_/U could form non-aqueous surfactant-free micro-emulsion through mixing with ethanol and toluene. The size of the micro-emulsion droplets was less than 7 nm and was dependent on the concentration of toluene. This newly prepared TrpBF_4_/U-based micro-emulsion was applied to extract the extra-heavy oil from UHO. Results showed that the oil recovery was enhanced by at least 11% compared with that without the TrpBF4/U micro-emulsions. The mechanism of the oil–solid separation was mainly ascribed to the interfacial alteration at the oil and mineral surfaces by the TrpBF_4_/U, including lowering the interfacial tension and increasing solid wettability. Jinjian et al. [173] further puts forward the ionic-liquid-enhanced solvent extraction mechanism in the oil–solid separation processes of UHO. As shown in Figure 18, the ionic liquids effect could be summarized for two aspects, namely, the heavy oil internal property alteration and interface property alteration. On the one hand, ionic liquids could alter the heavy oil internal property. The ionic liquids could increase the heavy oil C/H ratio, decrease the heavy oil viscosity, and increase the heavy components (asphaltenes, resins) content. On the other hand, ionic liquids could effectively alter the interface property (oil-solid interface and oil-water interface). Ionic liquids could increase the rock surface hydrophilicity and decrease the oil-water interfacial tension. In addition, the ionic liquids would alter the surface charge of rock surface and heavy oil surface. The ionic liquids can further effectively decrease the heavy oil and the CaCO_3_ charges absolute value, and then the heavy oil and CaCO_3_ surface attractive force would decrease. Therefore, the oil solid interface force would decrease. Xingang Li et al. [42] used a switchable solvent with a diamine structure called N, N, N′, N′-tetraethyl-1, 3-propanediamine (TEPDA) to extract heavy oil from UHO. This solvent can switch between its hydrophilic and hydrophobic states with the addition or removal of CO_2_, respectively. It founds that TEPDA can be applied as an organic solvent and can be ionized in water to form cations. TEPDA was proven to perform well with water in separating heavy oil from a carbonate solid surface, resulting in over 10% additional oil recovery. A mechanistic study showed that in this aqueous/non-aqueous hybrid process, TEPDA acts mainly as a solvent in softening and dissolving heavy hydrocarbons (dissolution effect). Meanwhile, a small amount of ionized TEPDA in the aqueous phase acts as a surface-active material accumulating at the oil-water and water-solid interfaces (interfacial modification). In addition, Yi Lu et al. [25] used CO_2_-responsive surfactants to enhance oil–solid separation in UHO. The mechanistic study showed that CO_2_-responsive surfactants were investigated to enhance the overall heavy oil recovery by switching their interfacial activity to the desired state in each stage. The surfactants at interfacially active state greatly enhanced the separation of heavy oil from hosting solids, as demonstrated by measuring contact angle and oil liberation using a custom-designed on-line visualization system (see Figure 19). 

### 3.2. Oil–Water Separation Mechanism

In the separation processes of UHO, oil–water separation (or demulsification) is the follow-up procedures of oil–solid separation. Chemical demulsification is the most common method in the exploitation and production of heavy oil [174,175,176,177,178]. The key to this method is to use chemical demulsifier with strong interfacial activity to realize oil–water separation. Therefore, understanding the mechanism of oil–water separation is the premise to realize the efficient demulsification of heavy oil–water emulsions. 

At present, the chemical demulsification principles with high recognition mainly include the following principles. (1) Replacement principle [179], in which the added chemical demulsifier possesses a stronger surface activity than the natural surfactants, and can preferentially adsorb on the oil–water interface and significantly reduce the oil–water interfacial tension, so as to replace the original film-forming materials at the oil–water interface and demulsify the emulsions. (2) The principle of antiphase action [180], in which, after adding some chemical demulsifiers (inverse demulsifiers) into the oil–water emulsions, these demulsifiers will interact with the natural surfactants in heavy oil to form complexes, which will lead to the transformation of emulsions (such as the transformation from W/O type to O/W type). In the process of transformation, the water phase settles under the action of gravity, so as to realize the demulsification of emulsions. (3) The wetting and solubilization principle [181], in which the demulsifier added to the emulsions can exist in the form of micelles, and the formed micelles have the effect of solubilizing the oil–water interfacial films. The film is destroyed by dissolution, so as to realize demulsification of oil–water emulsions. (4) The principle of neutralizing interface charge [182], in which the surface of oil–water emulsions possess negative charge. The cationic demulsifiers added can neutralize the interface charge, reduce the repulsive force between droplets, and make emulsion droplets coalesce, further realizing the demulsification of oil–water emulsions. 

A large number of studies have shown that the demulsification principle of heavy oil–water emulsions is mainly the interaction mechanism between demulsifiers and asphaltene interfacial films at the heavy oil–water interface [183,184,185,186,187]. For example, Zuoli Li et al. [188] studied the interactions of random biopolymer demulsifiers and heavy oil at oil–water interface for heavy oil dehydration. It was found that demulsifiers can result in less asphaltenes with rigid films, as indicated by the way that lower compression energies and lower collapse pressures correspond to a lower coalescence time of two water droplets. It appears that softening the asphaltenes interfacial films through the addition of polymeric demulsifiers promotes coalescence of two water droplets. Additionally, the water removal was inversely proportional to the coalescence time of the water droplets which was in turn impacted by the rigidity of the water–oil interfacial films. Wei Liu et al. [189] explored the interaction between asphaltene and hydrolyzed polyacrylamide (HPAM) demulsifier at the oil–water interface (Figure 20). Their results show that the addition of HPAM could change the emulsion type from W/O to O/W, and the shear modulus of the emulsions stabilized by HPAM and asphaltenes were ten-times lower than that of asphaltene emulsions. The interaction of amide and carboxylic acid groups in HPAM molecules with asphaltenes has been investigated through interfacial properties. The carboxylic groups and amide groups of HPAM molecules can interact with the interfacial active components of asphaltenes and form a carboxyl–asphaltene–amide complex unit, which make the asphaltene molecules “anchorage” on the HPAM molecular chain. Fan Ye et al. [187] revealed the demulsification mechanism of amine-functionalized recycling wastepaper demulsifier (AWP) in heavy oil–water emulsions. After the amphipathic and interfacial active AWP is added to emulsions, it quickly migrates to the oil–water interface and adsorbs asphaltene. Whereafter, AWP squeezes asphaltenes and softens and/or destroys the interfacial films. In addition, the negative charge of oil droplets in emulsions will also be electrostatically neutralized by the positive charge on the surface of AWP, thereby reducing the electrostatic repulsion of adjacent droplets. After that, the adjacent droplets coalesce into a big droplet and then aggregate into an oil floccule until the buoyancy of the oil floccule drives it to move to the top of the water. Zhongwei Li et al. [29] researched the demulsification mechanism of tannic acid-based polyether demulsifier in heavy oil–water emulsions. As shown in Figure 21. It was found that asphaltenes film was drained away with the strong interfacial interaction between demulsifier molecules and surface-active components. These components are composed of asphaltene, polymer and other subsequently added functional reagents. When water droplets approach together, their aggregation is helped by the demulsifier molecules. Coalescence easily occurred due to the weakened protection of these components, leading to the formation of large droplets. Their size remarkably increased to a maximum after 20 min accompanied by a sharply reduced number of droplets. As a result of the sedimentation of large droplets, the separated water at the bottom increased until the small amount of water droplets became hard to agglomerate. 

## 4. Challenges and Perspectives

As an important part of petroleum resources, the rational development of UHO possesses important industrial application value. At present, the main methods of separating UHO are hot water-based extraction (HWBE), solvent extraction, reaction extraction etc. [20,190,191,192]. There are still many key problems to be further solved in the oil–solid separation and oil–water separation processes involved in the UHO industrial process. Therefore, it is pivotal to understand the challenges related to oil–solid separation and oil–water separation, both theoretical and practical. The challenges and future researched developments of oil–solid separation and oil–water separation are discussed in detail below.

### 4.1. Construction of Oil–Solid Interaction Mechanism System

As explained earlier, the interaction between heavy oil and solid affects the separation effect of UHO. Therefore, a deep understanding the mechanism of oil–solid interaction is the basis for the efficient separation of UHO. 

At present, the research on the mechanism of oil–solid interaction has mainly focused on the interaction between heavy oil fractions (such as saturates, aromatics, resins, asphaltenes) and mineral components in heavy oil [193,194,195,196,197]. For example, Xingang Li et al. [198] systematically investigated the liberation of asphaltenes on the muscovite (KAl_2_(Si_3_Al)O_10_(OH)_2_) surface through instrumental characterization and molecular dynamics (MD) simulation in the separation of UHO (see in Figure 22). Micro force measurements by atomic force microscopy show that the adhesion force between asphaltenes and muscovite is weaker than that between asphaltenes and silica in both air and water. Assisted by the MD simulation, it was found that the detachment of asphaltenes is highly dependent upon the mineral types and the presence of the water film on the mineral surfaces. Although the van der Waals force is found to be the main force between asphaltenes and mineral surfaces, the presence of potassium ions (K^+^) on the muscovite surface could increase the percentage of the electrostatic forces in the total force. Furthermore, the presence of a 0.4 nm water layer (in the air) between asphaltenes and the muscovite surface can reduce their interactions dramatically compared to those in a vacuum state. Xingang Li et al. [199] have further studied the desorption behaviors of saturates, aromatics, resins, asphaltenes and bitumen on mineral solid particles (silica, kaolinite and calcium carbonate), as shown in Figure 23. They demonstrated that asphaltenes exhibit less desorption compared with those saturates and aromatics due to their stronger affinity to the mineral surfaces through polar and chemical interactions. This poor desorption of the heavier fractions caused more significant wettability alteration to the mineral surfaces and dominated the determination of the physicochemical and desorption properties of heavier fractions. Yun Bai et al. [200] further studied the effects of the N, O, and S heteroatoms on the adsorption and desorption of asphaltenes on silica surface through molecular dynamics simulation (MD). Their results reveal that the characteristic adsorption configuration of asphaltenes is ascribed to the competition between the asphaltene–silica interaction and π–π stacking interaction among the asphaltene polyaromatic rings. The presence of heteroatoms is found to be able to strengthen the interactions between asphaltenes and silica, depending on their type and location. For example, the terminal polar groups, especially the carboxyl (COOH), exhibit the greatest contribution to the electrostatic interaction (increasing from −81 to −727 kJ/mol). The S atoms are also found to increase the van der Waals interaction energies by 25%. According to the equilibrium desorption conformation and density profile, the presence of heteroatoms is found to significantly hinder the desorption of asphaltenes from silica due to the enhanced polar interactions. The impeded desorption is also confirmed by the slower detachment of asphaltenes based on the time-dependent interaction energies and center of mass (COM) distances analysis (see Figure 24). 

To sum up, some progress has been made on the heavy oil–solid interaction mechanism. However, further systematic research is still needed. In actual UHO, the heavy oil composition is extremely complex, and the physical and chemical characteristics of heavy oil components have an important impact on the oil–solid separation. As is known to all, heavy oil components in UHO are divided into saturates (S), aromatics (A), resins (R) and asphaltenes (A), and yet SARA are only roughly classified according to the polarity from weak to strong. The essence of each component is still a complex mixture containing multiple compositions and different molecular structures. Therefore, in the follow-up research process we should, on the one hand, strengthen the molecular structure analysis of SARA and build the interface system between SARA’s molecular structure and oil–solid interaction. On the other hand, a quantitative structure–activity relationship database should be established to correlate the three-level relationships of oil–solid structures, physicochemical properties and adsorption–desorption. This should include mineral composition/heavy oil composition–surface characteristics of mineral/heavy oil components (mineral surface potential, hydrophilicity, element composition, viscosity, density, etc.) and adsorption and desorption characteristics of heavy oil components on mineral surface. Furthermore, the oil–solid interaction law should be systematically explored at a deep level and the quantitative structure–activity relationship database gradually supplemented. Finally, on the basis of the above research, there should be the systematic construction of an oil–solid interaction mechanism system.

### 4.2. Construction of Oil–Water Interaction Mechanism System

In the process of separation of various unconventional oil ores (HWBE, SE, RE), the participation of water is involved in different degrees. However, because of the existence of interfacial active components and nano mineral particles in heavy oil, the unconventional oil ores often form stable heavy oil–water emulsions during the separation process, affecting subsequent separation, purification, storage and transportation [201,202,203]. Therefore, the demulsification of heavy oil–water emulsions is required. However, due to the complexity of the heavy oil composition, it is necessary to clarify the interaction mechanism between heavy oil and water, it being the theoretical basis for the realization of efficient demulsification of heavy oil–water emulsions. 

At present, the research on the interaction mechanism of heavy oil–water emulsions has mainly focused on the role of asphaltene components at the heavy oil–water interface. For example, as shown in Figure 25. Ramin Moghadasi et al. [204] investigated the mechanism of the surface behavior of asphaltenes in the presence of water. Their results indicate that aging reduces asphaltenes surface activity only when the crude oil is diluted, revealing that asphaltene precipitation is a time-dependent phenomenon. Finally, it identified that sudden pressure shocks have no significant effect on asphaltenes behavior in crude oil. However, related studies have shown that interfacially active asphaltenes (IAA) subfractions in asphaltenes are the most active subfractions at the oil–water interface [205,206]. Peiqi Qiao et al. [207] studied the influence of the solvent aromaticity on the interfacial characteristics of IAA. They found that IAA subfractions were considered to be irreversibly adsorbed at oil–water interfaces, differences in their elemental compositions led to differences in asphaltene interfacial activity and the properties of their interfacial films. The Toluene–IAA subfraction formed a densely packed network that was more elastic and resistant to drop–drop coalescence than the system prepared using Heptane–IAA. As indicated by FTIR and XPS analysis, the oxygenated groups, in particular sulfoxides, play a critical role in the interfacial activity of asphaltenes. On this basis, we combined experimental analysis and molecular dynamics simulation to analyze a monomer molecular structure model of IAA, and then explored its interfacial mechanism at the oil–water interface [208]. The results show that the nonpolar fatty chains in IAA molecules are intertwined. It was found that the π–π stacking force presents between different aromatic rings in IAA molecules (see Figure 26). In this way, these IAA molecules can self-aggregate at the oil–water interface (see Figure 27). In addition, due to the presence of heteroatoms (N,O,S), the IAA molecules would form hydrogen bonds with themselves and water molecules (see Figure 28). These non-covalent bonds allow the IAA molecules to form certain interfacial films at the oil–water interface. 

To summarize, at present, the research on heavy oil–water interaction mainly focuses on the interfacial films formed by asphaltene components at the oil–water interface. However, the molecular structure of the four components (SARA) in heavy oil is extremely complex. Follow-up research should focus on the analysis of various molecular structures of the SARA, establish a molecular structure model database, and systematically study the influence of the interface characteristics of the SARA on the characteristics of the heavy oil–water interfacial films. Finally, the heavy oil–water interaction mechanism system will be constructed.

### 4.3. Systematic Construction of Separation Mechanism of UHO

On the basis of understanding the mechanism of the oil–solid interaction and oil–water interaction in UHO industrial processes. The introduction of some new green extractants, such as ionic liquids [209], switchable solvents [24], and nanofluids [210], can enhance the oil–solid separation process, and further realize the efficient recovery of heavy oil from UHO. However, due to the participation of water and natural surfactants in UHO, extremely stable heavy oil–water emulsions are inevitably produced. Therefore, in combination with the previous research and the actual demand of industrial application prospects, it is necessary to introduce novel demulsifiers, such as esterified polyether [21], carboxylated polyether [33] and novel magnetic nanofluids [211,212], to realize high efficiency demulsification of heavy oil–water emulsions (see Figure 29). In addition, the systematic understanding of the separation mechanism of UHO is a prerequisite for developing new extractants or demulsifiers. However, the oil–solid separation process and the oil–water separation process of heavy oil mines are two cascaded processes, and the separation mechanism of UHO needs to be systematically constructed by combining the separation mechanism of these two processes, as shown in Figure 30. First of all, the molecular structure models of SARA in heavy oil should be analyzed by molecular structure characterization techniques such as gel permeation chromatograph (GPC), Fourier transform infrared spectrometer (FTIR), elemental analyzer (EA), nuclear magnetic resonance spectrometer (NMR), high resolution gas/liquid chromatography mass spectrometer, and then the multi-molecular structure models physical property database of heavy oil components should be established. Additionally, mineral composition and molecular structure characteristics of UHO should be analyzed by scanning electron microscope (SEM), FTIR, X-ray fluorescence spectrum analyzer (XRF), X-ray diffractometer (XRD), and X-ray photoelectron spectrometer (XPS). Then, the database of mineral molecular structure and physical properties should be established. Next, the interfacial characteristics of heavy oil components, extractants and demulsifiers should be analyzed by surface tensiometer, rheometer and viscometer. According to the interfacial characteristics, on the one hand, the oil–solid separation mechanism under the action of extractant is established from the atomic scale through quartz crystal microbalance (QCM), atomic force microscopy (AFM) and molecular dynamics simulation. On the other hand, based on the microscopic bonding interaction involved in the process of heavy oil–water separation (see Figure 31 and Figure 32) [208,213], combined with AFM, Langmuir–Blodgett (LB) membrane analyzer and multi-scale theoretical calculation, the separation mechanism of heavy oil–water can be established. Finally, the UHO separation mechanism should be structured systematically by combining the oil–solid separation mechanism and the oil–water separation mechanism. Thus, the solid theoretical foundation for the development of novel extractants and demulsifiers is laid out.

### 4.4. Research and Development of Novel Extractants and Demulsifiers

In the industrial process of UHO, the separation of heavy oil and solids requires the participation of extractants [214]. At present, some organic solvent extractants are widely used to separate heavy oil and solids from UHO. For example, Tong Wang et al. used organic solvents (alkanes, chloroform) to extract heavy oil from UHO [215]. They found that the recovery rate of heavy oil is less than 85%. Hong Sui et al. used a binary solvent of ethyl acetate and *n*-heptane and applied it together with ionic liquids (ILs, 1-ethyl-3-methyl imidazolium tetrafluoroborate ([Emim]BF_4_)) to extract heavy oil from UHO at ambient conditions [216]. Results of bottle tests show that the bitumen recovery is highly dependent on the volume ratio of ethyl acetate to *n*-heptane. The maximum recovery was obtained at the ethyl acetate-to-*n*-heptane ratio of 3:6. With external addition of ILs, an additional improvement of ∼10% of bitumen recovery was observed (from 83% to 93% at the ethyl acetate-to-*n*-heptane ratio of 3:6). In addition, other organic solvents (such as butane, pentane, octane, o-xylene, acetone, benzene et al.) are often used to separate heavy oil from UHO [217]. However, organic solvents possess obvious disadvantages that are not friendly to the environment. Therefore, it is necessary to develop green extractants. For example, Paula Berton et al. used IL trioctylammonium oleate ((HN_888_) (Oleate)) at a 1:3 IL/oil sand mass ratio, which can achieve bitumen extraction from high-grade UHO of ca. 100% with low solids content (<1%) in a fast, low-energy process [218]. The results demonstrate that the proper design of the IL can lead to efficient oil extraction without conventional solvents, without generating aqueous tailings, and with minimum energy consumption, that is, a production process with environmental impacts comparable to those associated with production of other hydrocarbon resources. Furthermore, as shown in Figure 33, previous studies of our group have shown that the switchable-hydrophilicity solvents (SHSs) also possess good application prospects in heavy oil–solid separation [219]. SHSs are a type of organic solvent with switchable miscibility between hydrophilicity and hydrophobicity [220,221]. These novel solvents are mainly nitrogen-containing compounds, including tertiary amines [222,223], secondary amines and amidines [224]. Generally, they possess high boiling points, meaning less volatility [225]. Due to the presence of amino groups, this type of chemical can react with CO_2_, switching between a water-immiscible state and a water-miscible salt. This switchablility suggests its potential in the application of heavy oil–solid separation. Therefore, in the follow-up study, combined with the switchable characteristics of some ionic liquids in CO_2_ [226,227], a novel green extractant composed of IL and SHSs can be further developed, and its responsive properties in CO_2_ can be used to realize the efficient separation of heavy oil-solid in the industrial process of UHO.

For the separation of heavy oil–water emulsions, chemical demulsification is widely used to break emulsions [228,229,230]. The key to this method is demulsifier. At present, different kinds of chemical demulsifiers are applied to demulsify heavy oil–water emulsions. For example, Abdelrahman O. Ezzat et al. [231] have investigated the high demulsification performance of two novel amphiphilic pyridinium ILs (ethoxylate amino pyridinium bromide IL, QEP; diquaternized pyridinium bromide IL, DQEP) in heavy oil–water emulsions. Their results show that the demulsification efficiencies of QEP and DQEP reached 100% for W/O emulsion (50/50 vol%) and (30/70 and 10/90 vol%), respectively, to confirm the ability of these pyridinium ILs to be applied as demulsifiers for heavy oil–water emulsions. Arafat Husain et al. [178] studied the demulsification performance of new pyridinium ionic liquids: 1-butyl-4-methylpyridinium tetrafluoroborate (BMPT), 1-butyl-4-methylpyridinium hexafluorophosphate (BMPH), and 1-butyl-4-methylpyridinium iodide (BMPI). Bottle test results revealed that BMPT, BMPH, and BMPI demulsifiers removed water from the heavy oil–water emulsions effectively, and the demulsification efficiency (% DE) increased with increasing dosage. BMPT, BMPH, and BMPI achieved the best % DE of 84%, 99%, and 59%, respectively, at 1000 ppm after 60 min. In addition, Zejun Zhang et al. [232] synthesized a novel GTE–DDA demulsifier with special three-branch structure by a simple one-step method using glycerol triglycidyl ether (GTE) as core and dodecyl amine (DDA) as branched chain. The results of their bottle test showed that GTE–DDA quickly demulsified the W/O emulsion and the optimal DE reached 87.55% after demulsifying with 400 mg/L of GTE–DDA at 50 °C for 30 min. However, due to their environmentally friendly properties, non-ionic demulsifiers have attracted increasing attention from industrial fields [233]. As shown in Table 1**,** some non-ionic demulsifiers have been used to demulsify extremely stable heavy oil–water emulsions. Although the above nonionic chemical demulsifiers can demulsify these heavy oil emulsions, their performance still needs to be improved in demulsification rate or demulsification efficiency. Long demulsification time, high concentration of demulsifier and high demulsification temperature would not only increase costs but also waste resources [234]. Therefore, in combination with our previous research results [33], a new type of green demulsifier (hydrophilic groups modified non-ionic polyether) should be developed for the efficient demulsification of stable heavy oil–water emulsions. These hydrophilic groups (such as hydroxyl group, carboxyl group, ester group, amino group) modified with non-ionic polyether can possess the following two significant advantages in the demulsification of heavy oil–water emulsions: (1) Good amphiphilicity and high interfacial activity, which is beneficial for the promotion of the diffusion of demulsifier molecules in the two heavy oil–water phases and the stable adsorption at the heavy oil–water interface. (2) Abundant oxygen-containing groups can realize rapid destruction of asphaltene stabilized heavy oil–water interfacial films, so as to achieve the fast separation of heavy oil-water two phases.

To sum up, the development of novel extractants and demulsifiers is very important for the oil–solid separation process and the oil–water separation process. However, there are many kinds of extractants and demulsifiers, and it takes a lot of time to select them only through experiments. Based on this, machine learning provides an effective scientific means to solve this problem. Relevant research has shown that machine learning possesses a significant application effect in material screening. For example, as shown in Figure 34. An Chen et al. generated a comprehensive dataset from the periodic table consisting of 1092 potential 2D semiconductor/metal heterojunctions with good contact performances and demonstrated that the small interfacial dipole and the elimination of localized surface states are essential for designing advanced 2D metal–semiconductor systems with small SBHs. Their results show that the proposed method takes less than five seconds to execute and is far superior to traditional first-principles calculations in both time and cost, demonstrating the superiority of using machine learning for screening materials [214]. Furthermore, Sicong Ma et al. [235] have described a strategy of using machine learning to predict the structure of different intermediates in heterogeneous catalytic reactions, and the bridge between catalyst structures and catalytic performance was built by machine learning (Figure 35). Therefore, based on the separation mechanism of UHO. We can use this prediction and screening method of machine learning to guide the screening of novel extractants and demulsifiers, and further realize the efficient separation of oil–solid and oil–water in the industrial process of UHO. In summary, the novel extractants and demulsifiers are the core of oil–solid separation and oil–water separation in the industrial process of UHO. Based on this core foundation, in the future, to systematically improve oil–solid and oil–water separation research for UHO, and facilitate the industrial separation efficiency of UHO, more efforts should be made to commence studies on the separation technologies of UHO.

**Table 1 ijms-24-00074-t001:** Demulsification of heavy oil–water emulsions by non-ionic demulsifiers. Reprinted with permission from Ref. [2]. Copyright 2022, copyright the Elsevier.

Demulsifier	Concentration (ppm)	Demulsification Time (min)	Temperature (℃)	Dehydration Ratio (%)	References
SD31	100	60	60	85	[236]
Biodegradable Ethylcellulose	145	60	80	90	[237]
TECA	250	360	60	40	[32]
Ethoxylated surfactant	1000	30	70	80	[238]
PEO/PPO (1)	100	300	40	98.48	[239]
TAPA	100	90	70	91.7	[29]
Star polymer	1500	180	60	100	[240]
PEO/PPO (2)	2500	60	60	<17	[241]

Notes: SD31 is nonionic amphiphilic block copolymers of dendrimer copolymer. TECA is the nonionic tri-etherified cardanoxy amine. Ethoxylated surfactant is based on polyethylenimine modified with nonionic surfactants. PEO/PPO (1) is nonionic poly(ethylene oxide) poly(propylene oxide) copolymer demulsifiers. TAPA is tannic acid-based polyether demulsifier. Star polymer is (2Z,2′Z,2″Z)-4,4′,4″-((nitrilotris(ethane-2,1-diyl))tris(oxy))tris(4-oxobut-2-enoic acid) PEO/PPO (2) is poly(ethylene oxide)/poly(propylene oxide) block copolymer.

## 5. Conclusions

As an alternative energy resource of huge reserves, unconventional heavy oil ores are attracting increased attention from people around the world. In the industrial separation process of UHO, due to the participation of water and solvents, the processes of oil–solid separation (OSS) and oil–water separation (OWS) are inevitable. This work has systematically reviewed the basic theories of OSS and OWS. For OSS processes, the oil–solid separation efficiency is determined by solid wettability, contact angle and oil–solid interactions. For OWS processes, the interfacial active substances in heavy oil promote the stability of heavy oil–water emulsions. Interfacially active asphaltenes play a particularly key role in stabilizing heavy oil–water emulsions by forming strong interfacial films at the oil–water interface. These interfacial films hinder the coalescence of water droplets in emulsions, stabilizing them. In addition, the corresponding OSS and OWS mechanisms have been discussed. However, research on OSS and OWS is still in its fragmentary stages. There are still some gaps or challenges for OSS and OWS processes, and we put forward corresponding resolution strategies from the aspects of the construction of an oil–solid interaction mechanism system, the construction of an oil–water interaction mechanism system, the construction of a separation mechanism system, and the research and development of novel extractants and demulsifiers. Additionally, this review may even be useful in providing a framework of research prospects which may guide future research directions in laboratories and industries that focus on the OSS and OWS processes. It would also be of great interest to widespread interdisciplinary researchers (chemists, physicists and material scientists) who are working on interfaces, surfaces, colloids and energy. Last, but not least, future research should not only emphasize the improvement of OSS and OWS processes, but also pay more attention to the engineering considerations and development of novel extractants and demulsifiers for industrial application.

## Figures and Tables

**Figure 1 ijms-24-00074-f001:**
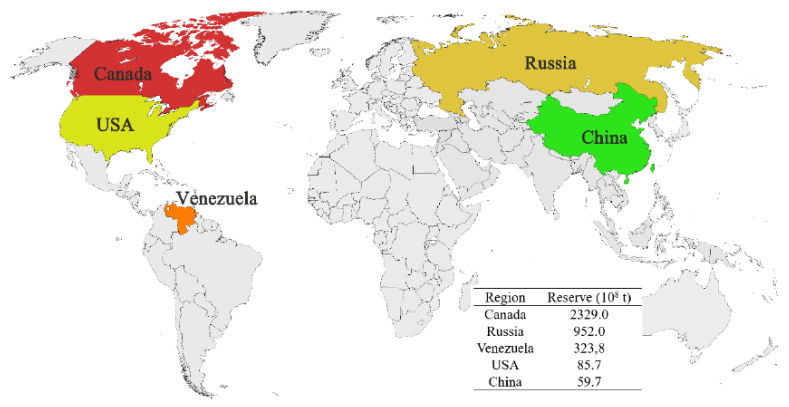
Distribution and reserves of unconventional petroleum oil resources. Reprinted with permission from Ref. [6]. Copyright 2011, copyright World Petroleum Congress. More details on Copyright and Liscensing are available via the following link: https://marketplace.copyright.com/.

**Figure 2 ijms-24-00074-f002:**
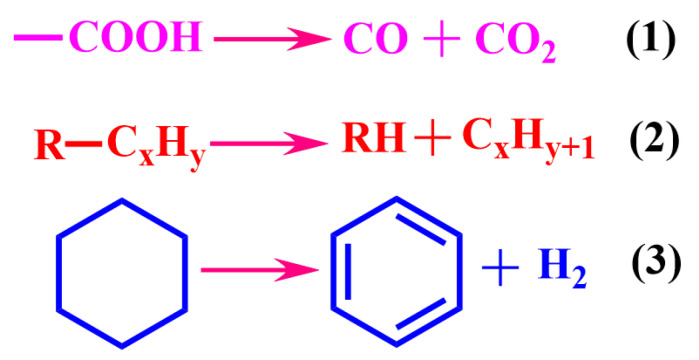
Pyrolysis process of heavy oil. Reprinted with permission from Ref. [16]. Copyright 2017, copyright Elsevier.

**Figure 3 ijms-24-00074-f003:**
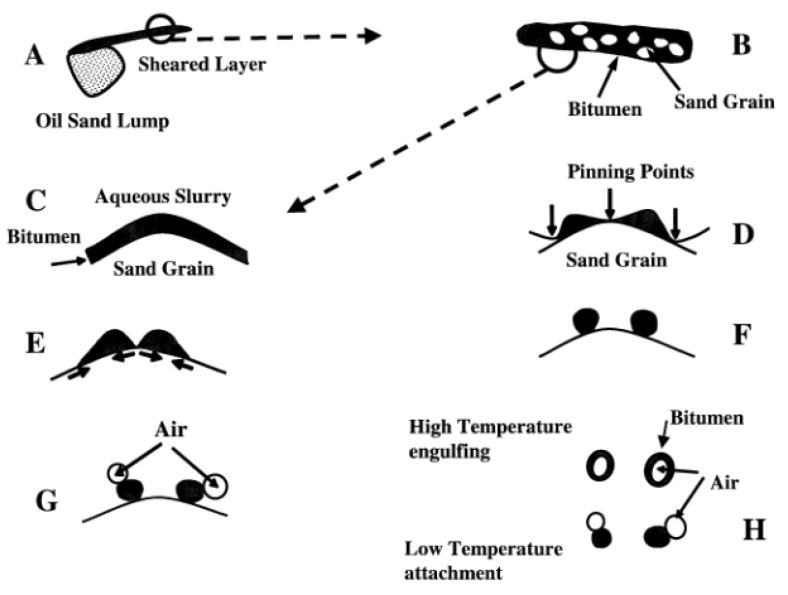
The separation processes of unconventional heavy oil ores by HWBE. Reprinted with permission from Ref. [18]. Copyright 2008, copyright Wiley. More details on Copyright and Liscensing are available via the following link: https://onlinelibrary.wiley.com/.

**Figure 4 ijms-24-00074-f004:**
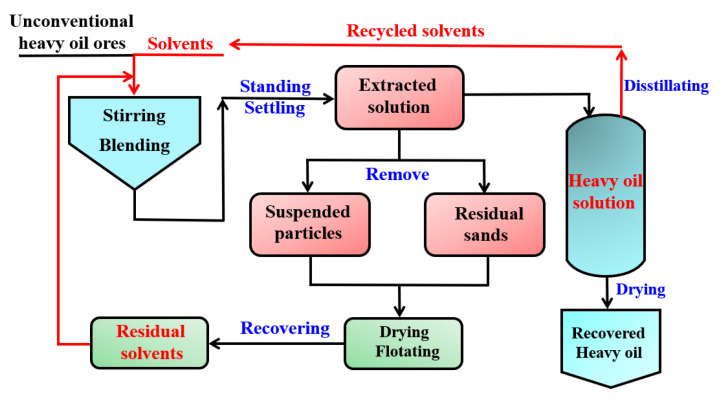
Schematic of the solvent extraction process for recovering heavy oil.

**Figure 5 ijms-24-00074-f005:**
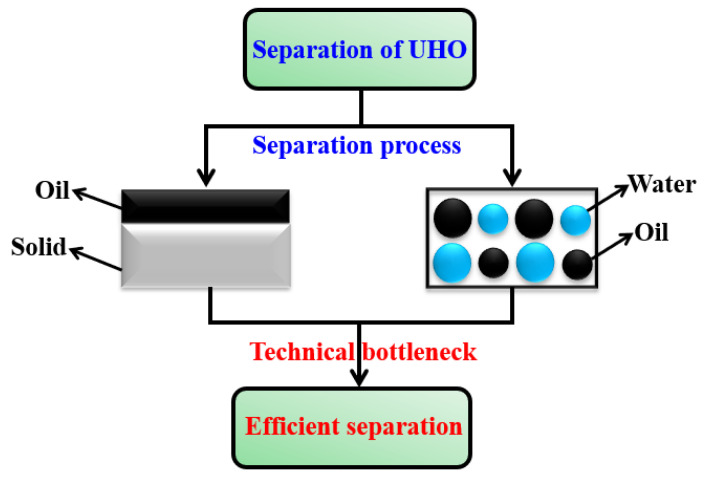
The key technical bottleneck of oil–solid separation and oil–water separation process.

**Figure 6 ijms-24-00074-f006:**
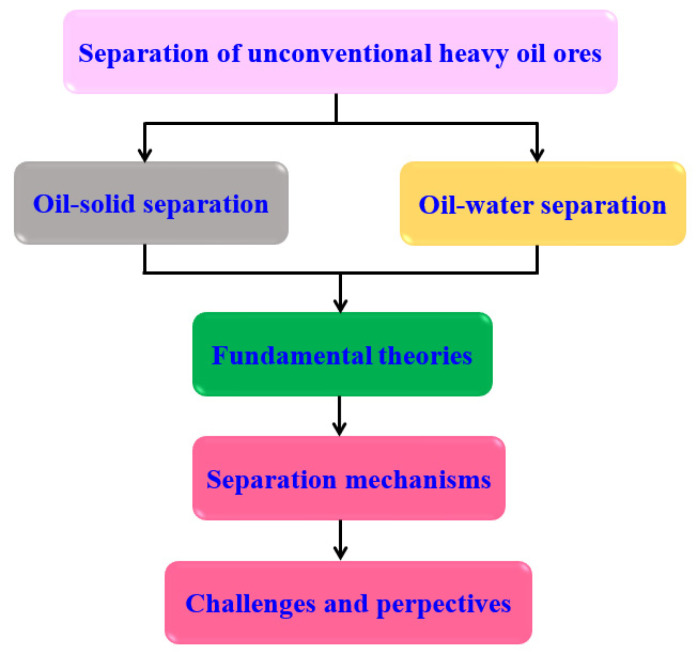
The review logical structure.

**Figure 7 ijms-24-00074-f007:**
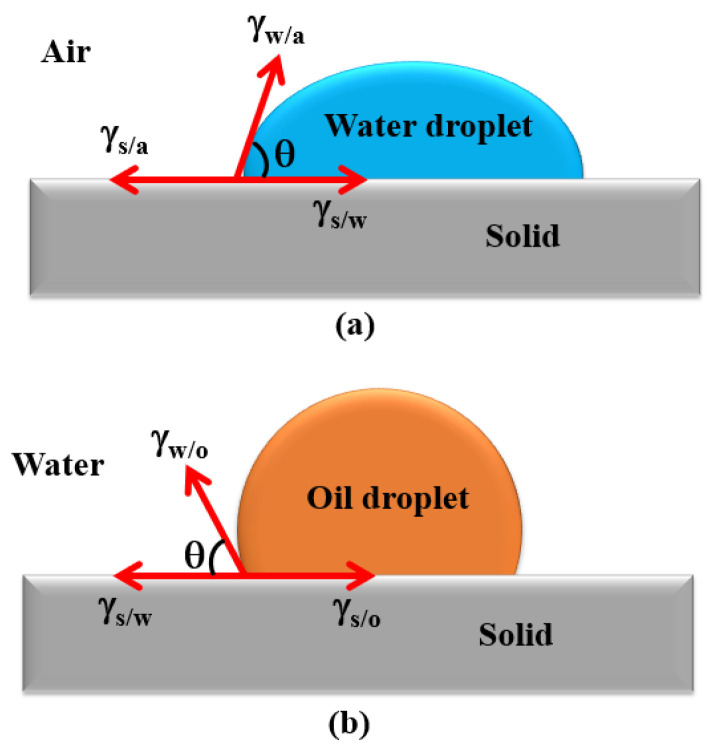
Schematics of contact angle for two systems: (**a**) solid–water–air system and (**b**) solid–oil–water system.

**Figure 8 ijms-24-00074-f008:**
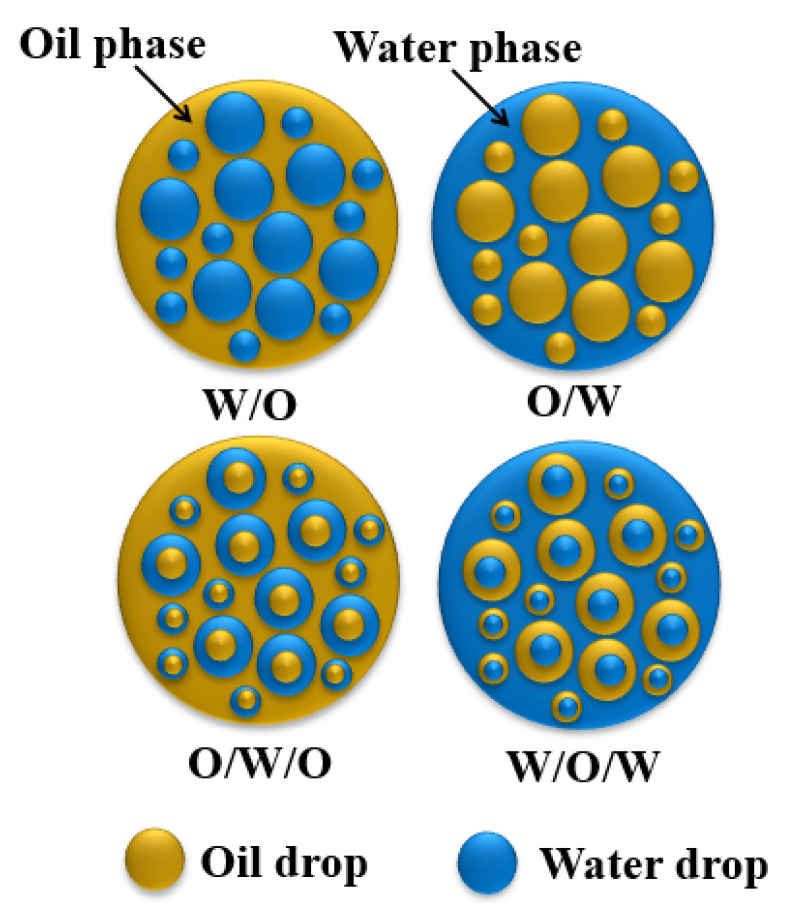
The types of heavy oil–water emulsions.

**Figure 9 ijms-24-00074-f009:**
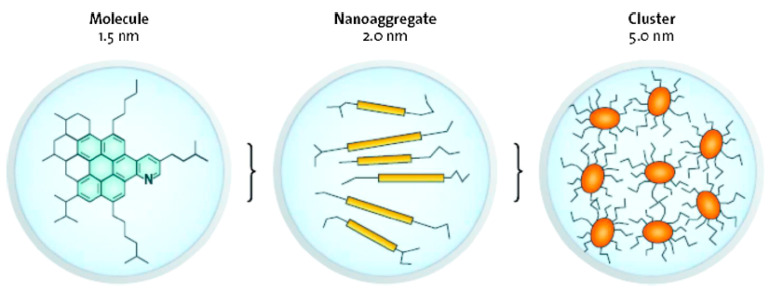
The molecular and colloidal structures of the Yen–Mullins model asphaltenes. Reprinted with permission from Ref. [92]. Copyright 2011, copyright Annual Reviews Inc. More details on Copyright and Liscensing are available via the following link: https://www.mendeley.com/.

**Figure 10 ijms-24-00074-f010:**
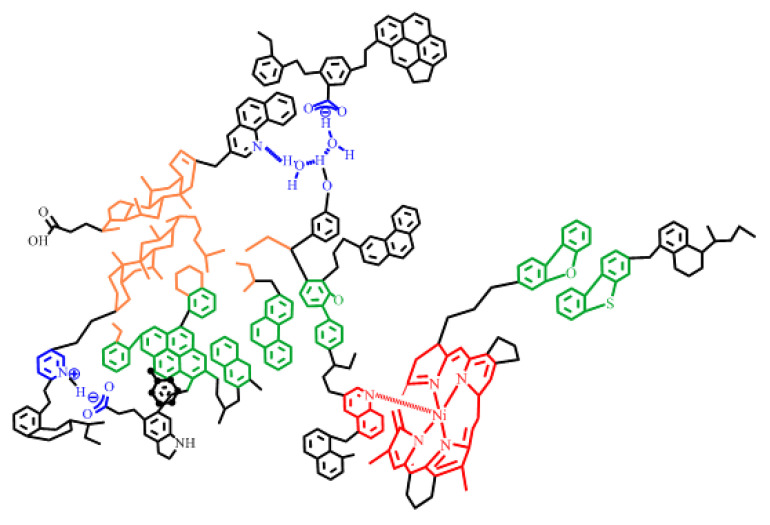
Proposed asphaltene molecular structure model. Reprinted with permission from Ref. [93]. Copyright 2011, copyright American Chemical Society. More details on Copyright and Liscensing are available via the following link: https://pubs.acs.org/.

**Figure 11 ijms-24-00074-f011:**
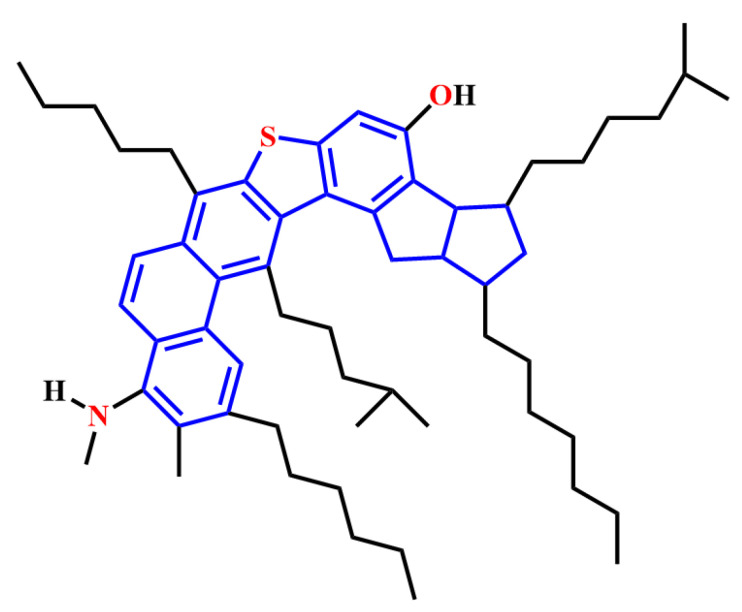
Typical molecular structures of resins. Reprinted with permission from Ref. [100]. Copyright 1996, copyright the American Chemical Society.

**Figure 12 ijms-24-00074-f012:**
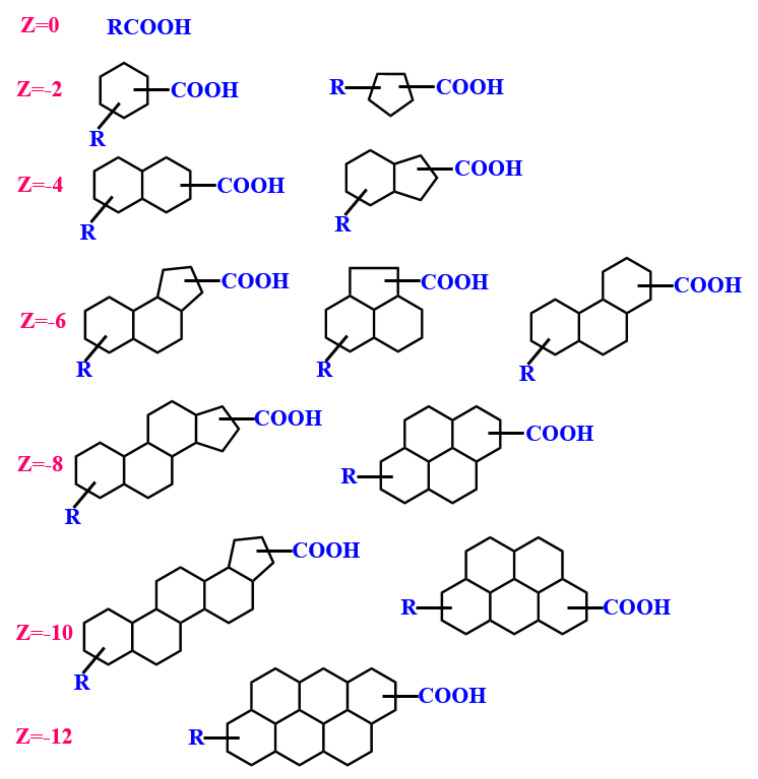
Typical structures of naphthenic acids with different carbon numbers and a varying degree of condensation, R represents the hydrocarbon tails. Reprinted with permission from Ref. [108]. Copyright 1991, copyright the American Chemical Society.

**Figure 13 ijms-24-00074-f013:**
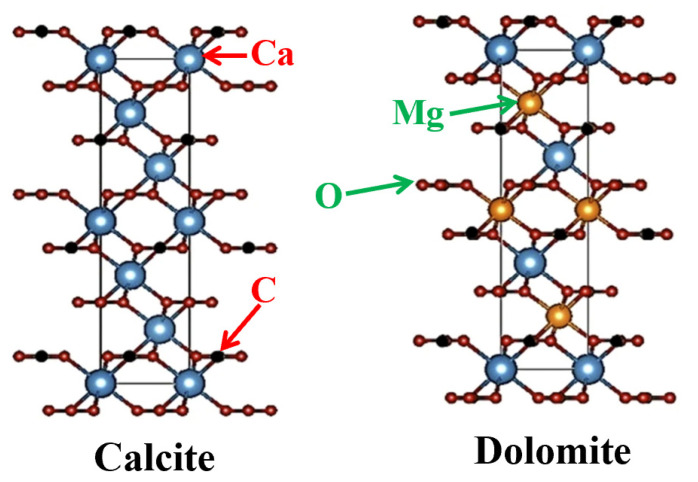
The structure models of carbonate and silicate. Reprinted with permission from Ref. [110]. Copyright 2019, copyright The Springer Nature. More details on Copyright and Liscensing are available via the following link: http://creativecommons.org/licenses/by/4.0/.

**Figure 14 ijms-24-00074-f014:**
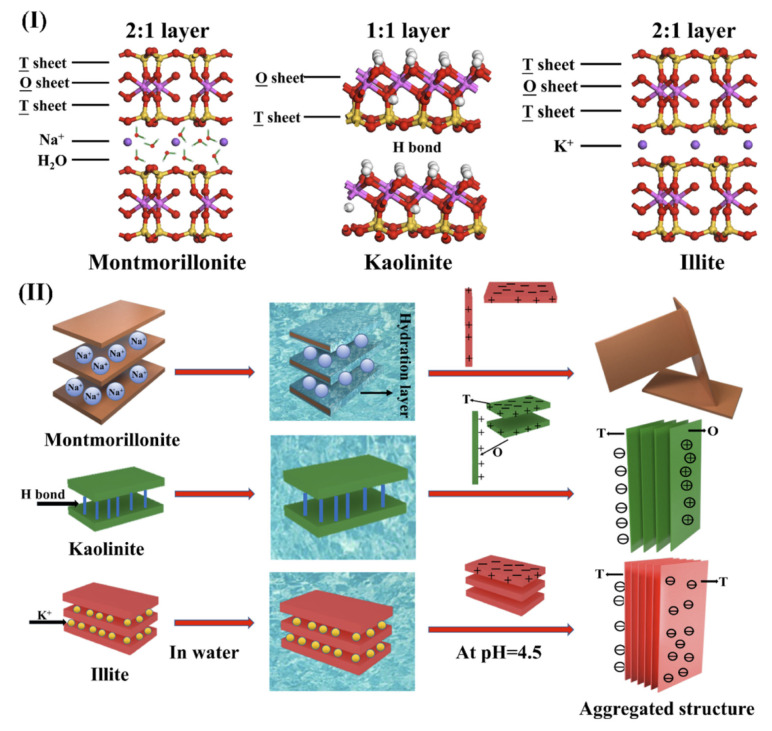
Schematic of the structures of clay minerals (**I**) and schematic illustration of different behaviors derived from various structures of clay minerals (**II**). Reprinted with permission from Ref. [116]. Copyright 2020, copyright the American Chemical Society. More details on Copyright and Liscensing are available via the following link: https://pubs.acs.org/.

**Figure 15 ijms-24-00074-f015:**
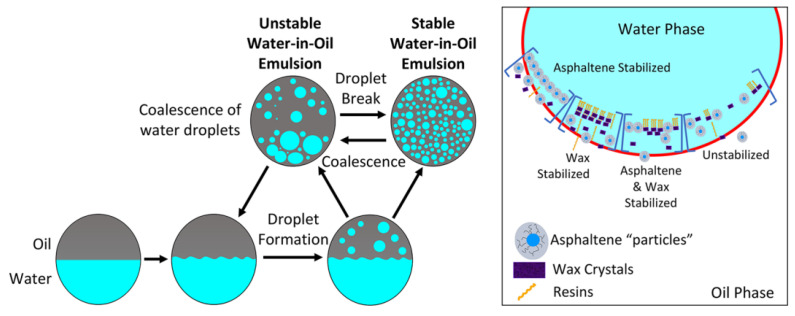
General mechanisms involved in the emulsification and demulsification processes of water-in-oil emulsions in the presence of asphaltenes. Reprinted with permission from Ref. [155]. Copyright 2022, copyright the American Chemical Society. More details on Copyright and Liscensing are available via the following link: https://pubs.acs.org/.

**Figure 16 ijms-24-00074-f016:**
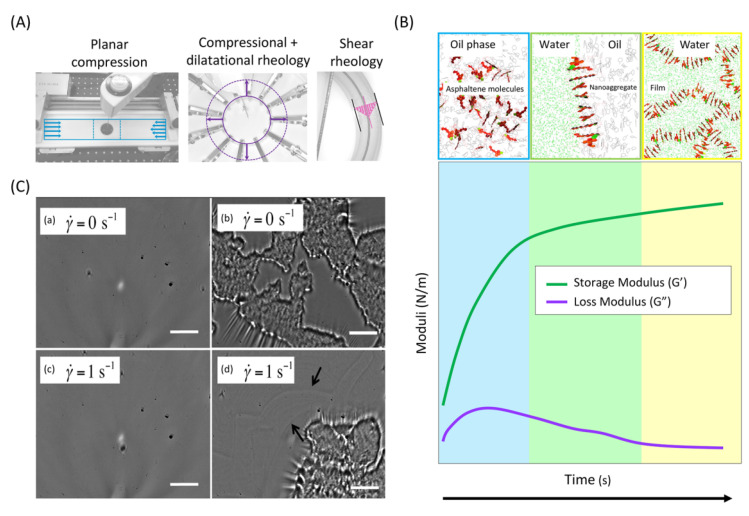
(**A**) Overview of compression, dilatational, and shear rheology techniques and the corresponding velocity profiles used to characterize asphaltene–stabilized interfacial films. Reprinted with permission from Ref. [170]. Copyright 2020, copyright American Chemical Society. (**B**) Representation of the formation of the asphaltene–stabilized interfacial film at the oil–water interface. Reprinted with permission from Ref. [89]. Copyright 2015, copyright American Chemical Society. Rheological response shows increasing elastic and decreasing viscous moduli. Adapted with permission from Ref. [171]. Copyright 2009, copyright EDP Sciences. (**C**) Optical microscopy images of asphaltene microstructure at air–water interfaces showing (a) uniform “skin” for low coverage at rest and (b) wrinkled “skin” with multilayer aggregates for high coverage at rest, (c) no changes for low coverage under shear rate, and (d) a fracture in the “skin” for high coverage caused by shear stress. Reprinted with permission from Ref. [172]. Copyright 2018, copyright The Society of Rheology.

**Figure 17 ijms-24-00074-f017:**
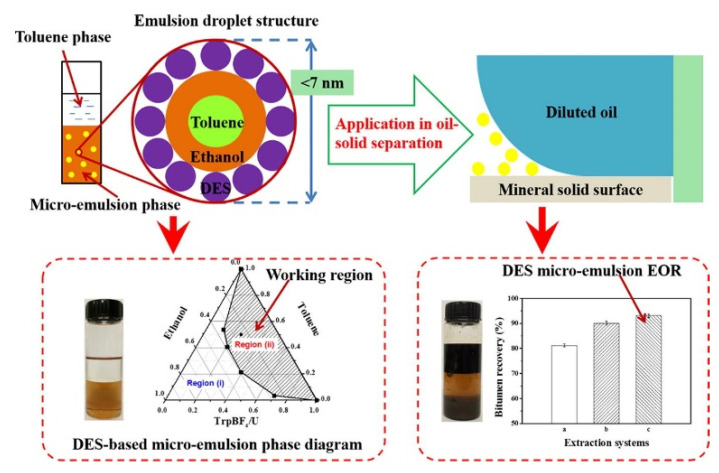
Application of amino acid ionic liquid-based deep eutectic solvent (TrpBF_4_/U) in UHO separation. Reprinted with permission from Ref. [43]. Copyright 2018, copyright the Elsevier. More details on Copyright and Liscensing are available via the following link: https://linkinghub.elsevier.com/.

**Figure 18 ijms-24-00074-f018:**
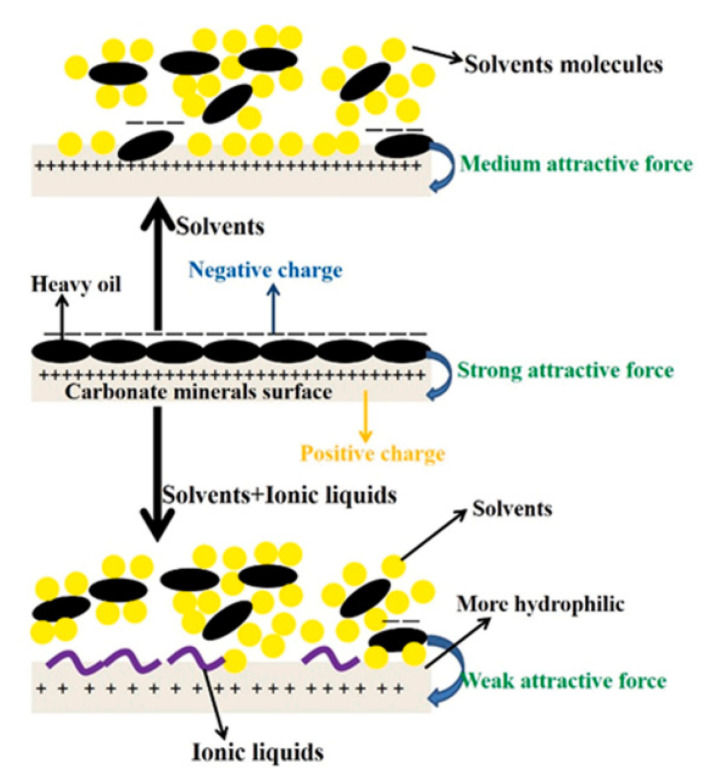
Ionic-liquid-enhanced solvent extraction mechanism. Reprinted with permission from Ref. [173]. Copyright 2022, copyright the Elsevier. More details on Copyright and Liscensing are available via the following link: https://www.sciencedirect.com/.

**Figure 19 ijms-24-00074-f019:**
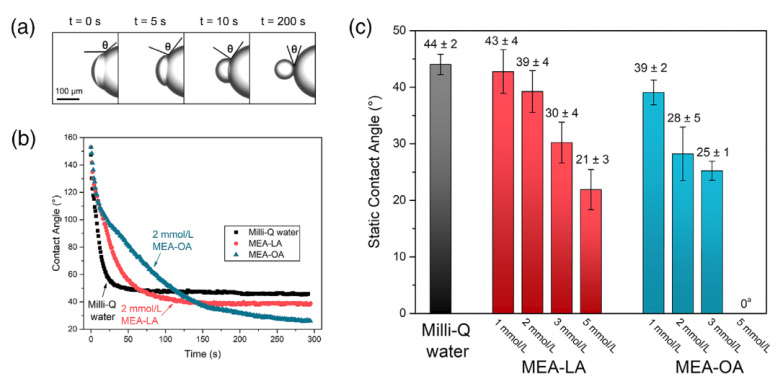
(**a**) Snapshots of a high viscosity oil droplet receding on a silica surface in Milli-Q water; (**b**) Dynamic contact angle in the first 300 s; and (**c**) Static contact angles in different aqueous solutions. ^a^ The oil drop detached from the silica surface in 5 mmol/L MEA-OA solution, thus showing as a contact angle of zero degree. Reprinted with permission from Ref. [25]. Copyrights 2021, copyright Wiley. More details on Copyright and Liscensing are available via the following link: https://aiche.onlinelibrary.wiley.com/.

**Figure 20 ijms-24-00074-f020:**
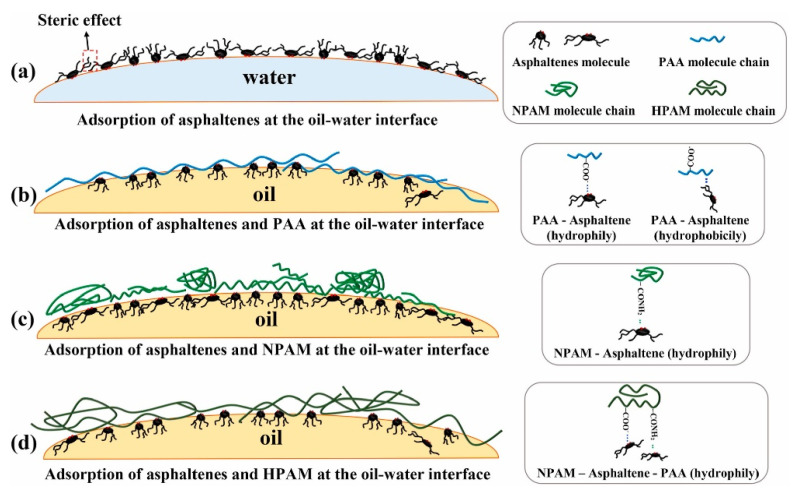
Schematic diagram of interaction on the oil–water interface. Formation of asphaltenes interfacial films (**a**). Interfacial films of synergistic interaction between PAA and asphaltenes (**b**). Interfacial films of synergistic interaction between NPAM and asphaltenes (**c**). Interfacial films of synergistic interaction between HPAM and asphaltenes (**d**). Reprinted with permission from Ref. [189]. Copyright 2022, copyright the Elsevier. More details on Copyright and Liscensing are available via the following link: https://www.sciencedirect.com/.

**Figure 21 ijms-24-00074-f021:**
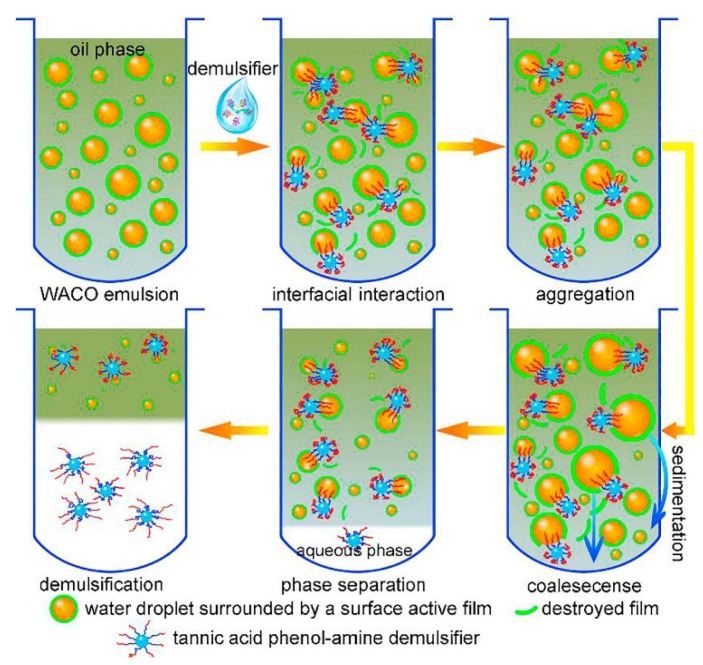
Schematic demulsification mechanism of tannic acid-based polyether demulsifier in heavy oil–water emulsions. Reprinted with permission from Ref. [29]. Copyright 2018, copyright the Elsevier. More details on Copyright and Liscensing are available via the following link: https://www.sciencedirect.com/.

**Figure 22 ijms-24-00074-f022:**
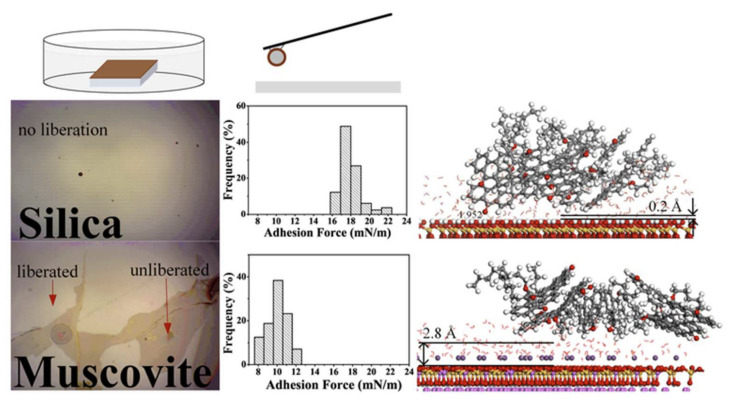
The liberation mechanism of asphaltenes on muscovite surface. Reprinted with permission from Ref. [198]. Copyright 2017, copyright the American Chemical Society. More details on Copyright and Liscensing are available via the following link: https://pubs.acs.org/.

**Figure 23 ijms-24-00074-f023:**
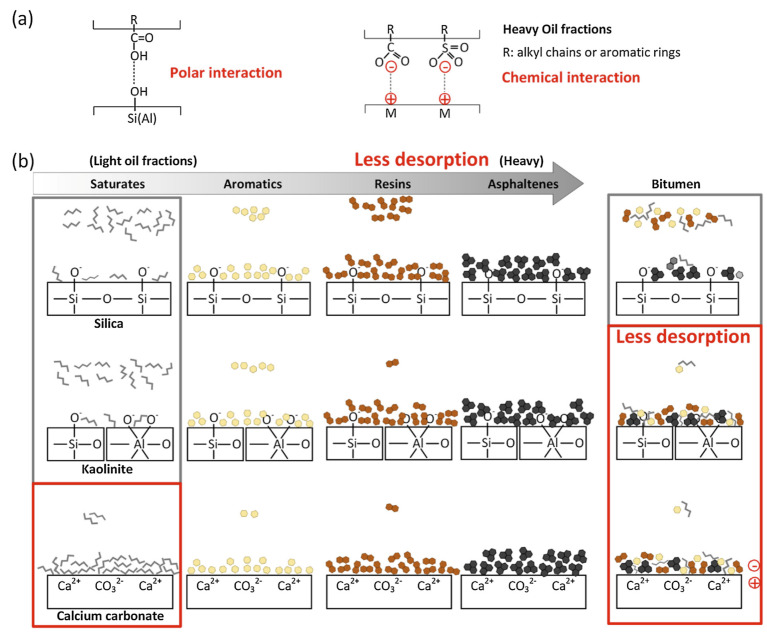
Schematics of (**a**) adsorption polar and chemical interactions between oil fractions and silica and kaolinite; (**b**) desorption properties of SARA and bitumen on silica, kaolinite and calcium carbonate in aqueous solutions. Reprinted with permission from Ref. [199]. Copyright 2018, copyright the Elsevier. More details on Copyright and Liscensing are available via the following link: https://linkinghub.elsevier.com/.

**Figure 24 ijms-24-00074-f024:**
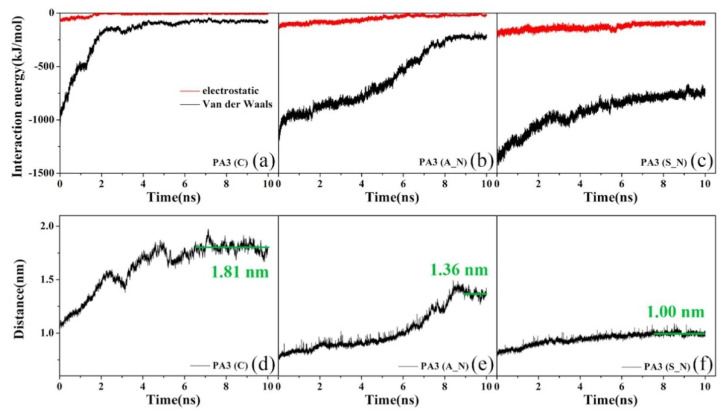
The interaction energies between asphaltenes (**a**) PA3 (C), (**b**) PA3 (A_N), (**c**) PA3 (S_N) and silica as a function of simulation time; average distance between center of mass (COM) of asphaltenes, (**d**) PA3 (C), (**e**) PA3 (A_N), (**f**) PA3 (S_N) and silica as a function of simulation time. Reprinted with permission from Ref. [200]. Copyright 2019, copyright the Elsevier. More details on Copyright and Liscensing are available via the following link: https://linkinghub.elsevier.com/.

**Figure 25 ijms-24-00074-f025:**
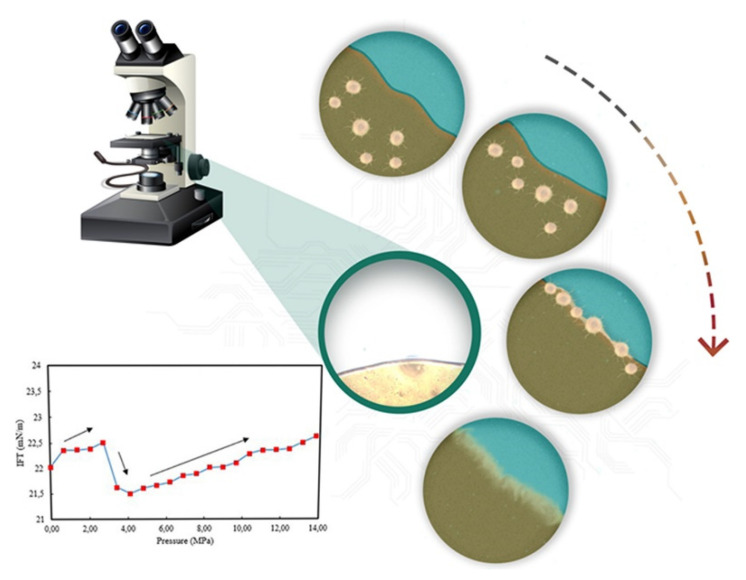
Asphaltenes surface behavior at oil–water interface. Reprinted with permission from Ref. [204]. Copyright 2019, copyright the Elsevier. More details on Copyright and Liscensing are available via the following link: https://www.sciencedirect.com/.

**Figure 26 ijms-24-00074-f026:**
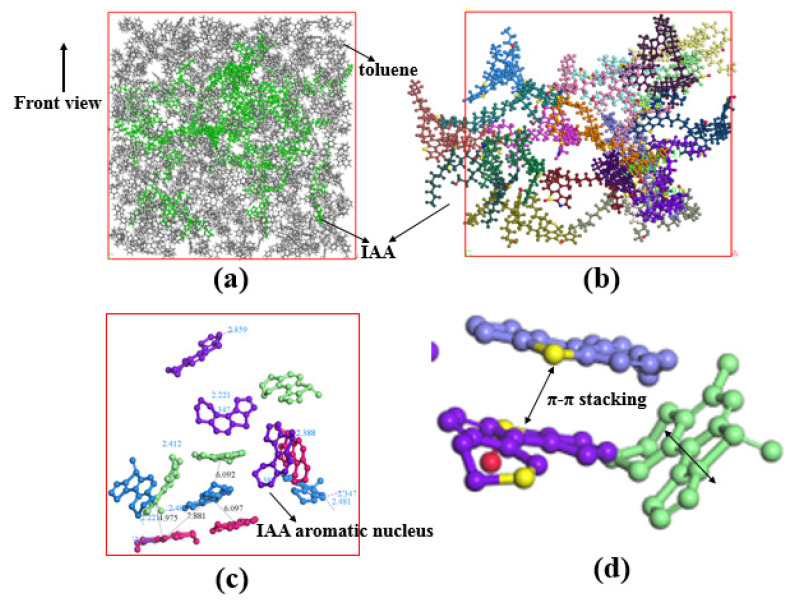
Aggregated system of IAA containing heteroatom ((**a**): initial configuration, (**b**): toluene molecules were hidden, (**c**): hydrogen atoms and fat chain were hidden (the blue number represents the hydrogen bond distance, the black number represents the π–π bond distance, unit is Å), (**d**): local amplification). Reprinted with permission from Ref. [208]. Copyright 2021, copyright with the Elsevier. More details on Copyright and Liscensing are available via the following link: https://www.sciencedirect.com/.

**Figure 27 ijms-24-00074-f027:**
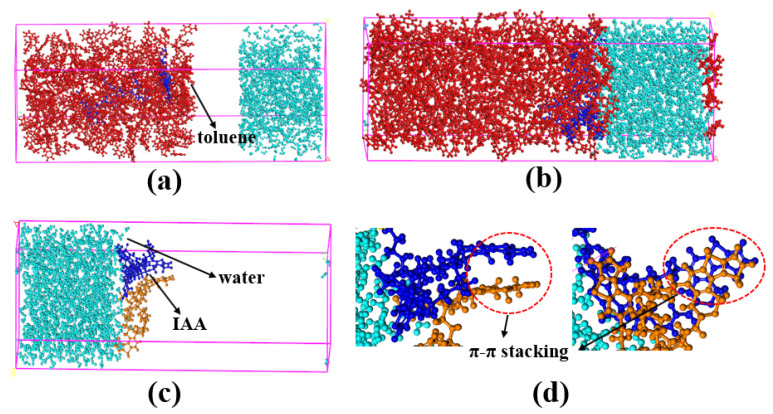
The adsorption of IAA at oil–water interface ((**a**): initial configuration, (**b**): final configuration, (**c**): toluene molecules are hidden, (**d**): local amplification). Reprinted with permission from Ref. [208]. Copyright 2021, copyright with the Elsevier. More details on Copyright and Liscensing are available via the following link: https://www.sciencedirect.com/.

**Figure 28 ijms-24-00074-f028:**
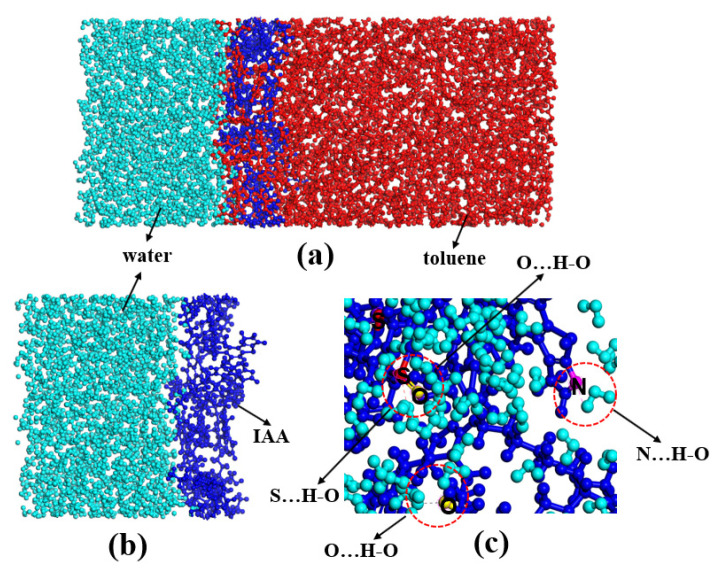
Interface system of IAA containing heteroatom ((**a**): initial configuration, (**b**): display hydrogen bonds, (**c**): Partial enlarged view). Reprinted with permission from Ref. [208]. Copyright 2021, copyright with the Elsevier. More details on Copyright and Liscensing are available via the following link: https://www.sciencedirect.com/.

**Figure 29 ijms-24-00074-f029:**
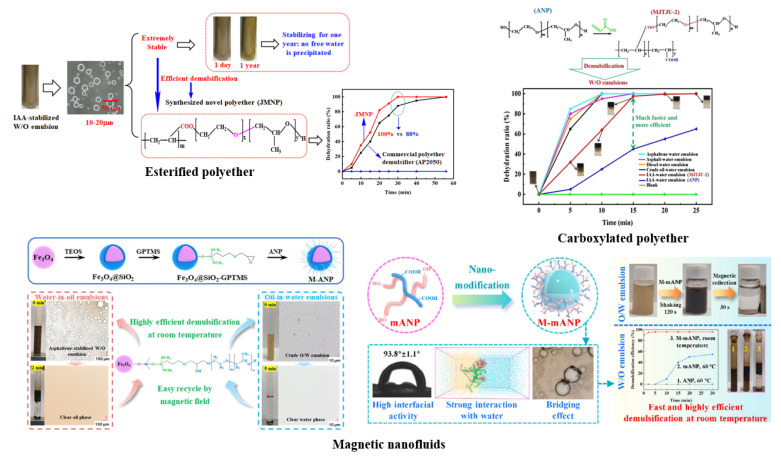
Application of new demulsifier in breaking heavy oil–water emulsions. Reprinted with permission from Refs. [21,33,211,212]. Copyright 2020, copyright the American Chemical Society. Copyright 2021 and 2022, copyright the Elsevier. More details on Copyright and Liscensing are available via the following links: https://pubs.acs.org/; https://www.sciencedirect.com/.

**Figure 30 ijms-24-00074-f030:**
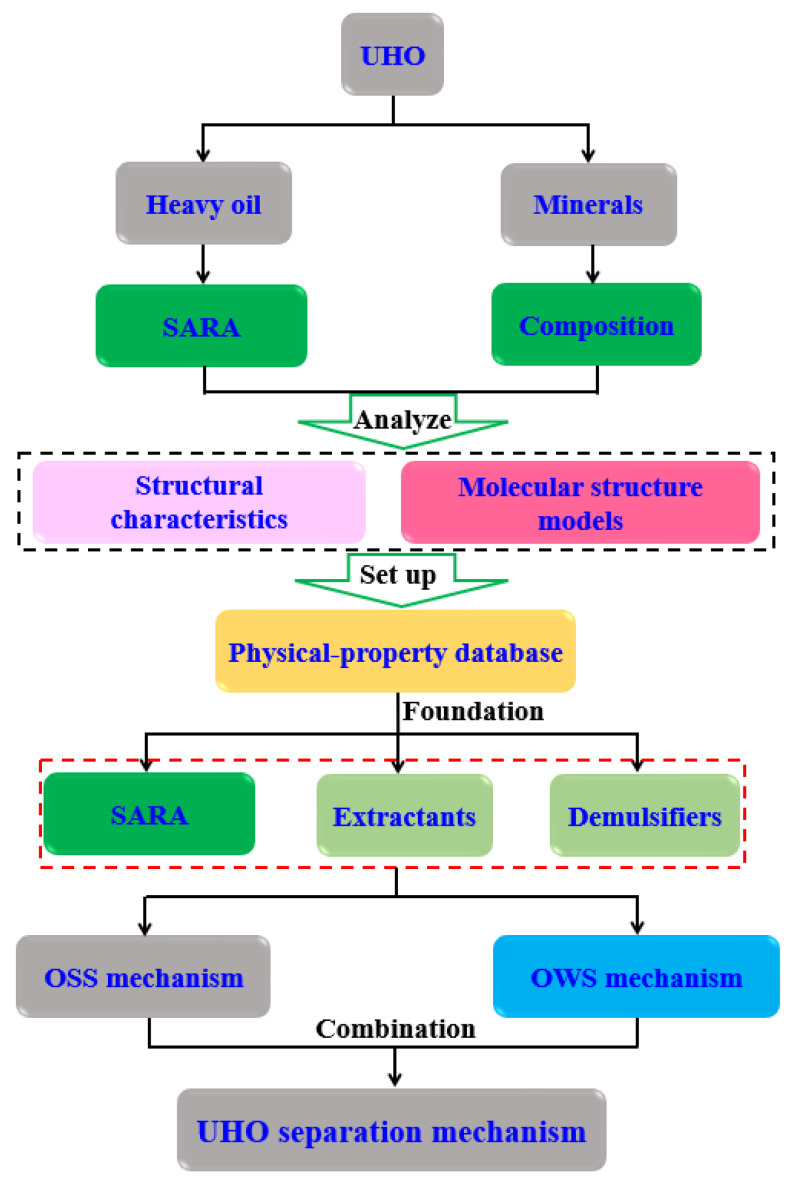
Schematic diagram of UHO separation mechanism construction.

**Figure 31 ijms-24-00074-f031:**
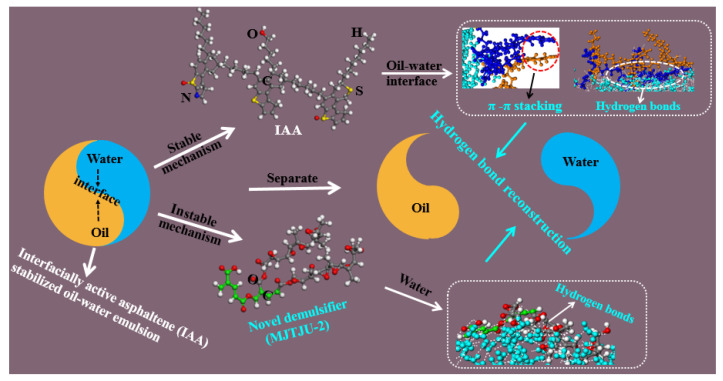
Hydrogen bond reconstruction in heavy oil–water separation. Reprinted with permission from Ref. [208]. Copyright 2021, copyright with the Elsevier. More details on Copyright and Liscensing are available via the following link: https://www.sciencedirect.com/.

**Figure 32 ijms-24-00074-f032:**
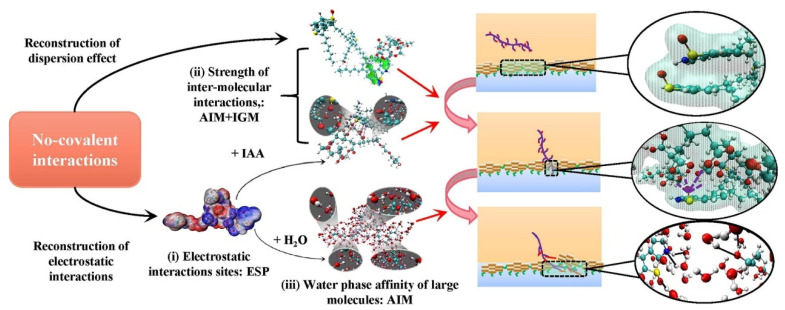
The non-covalent interactions in heavy oil–water separation. Reprinted with permission from Ref. [213]. Copyright 2022, copyright the Elsevier. More details on Copyright and Liscensing are available via the following link: https://www.sciencedirect.com/.

**Figure 33 ijms-24-00074-f033:**
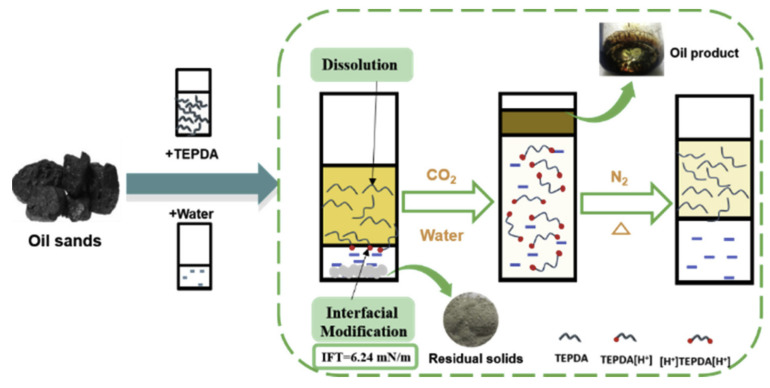
The schematics of the SHS–water hybrid process in separating bitumen from Indonesia carbonate asphalt rocks. Reprinted with permission from Ref. [219]. Copyright 2019, copyright the Elsevier. More details on Copyright and Liscensing are available via the following link: https://www.sciencedirect.com/.

**Figure 34 ijms-24-00074-f034:**
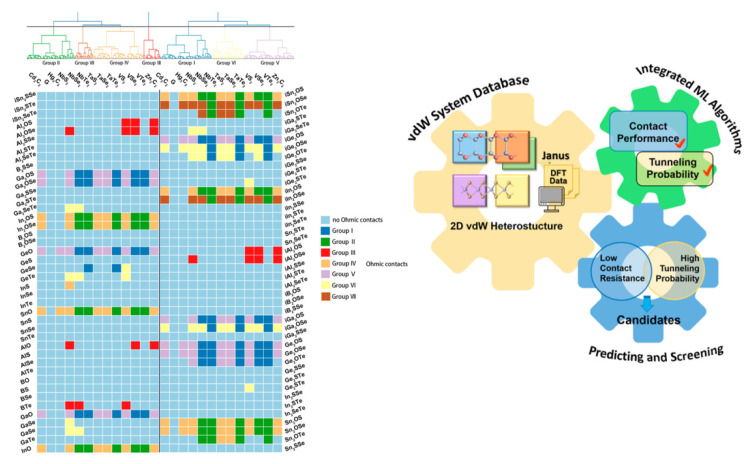
The application of machine learning in material screening. Reprinted with permission from Ref. [214]. Copyright 2022, copyright the American Chemical Society. More details on Copyright and Liscensing are available via the following link: https://pubs.acs.org/.

**Figure 35 ijms-24-00074-f035:**
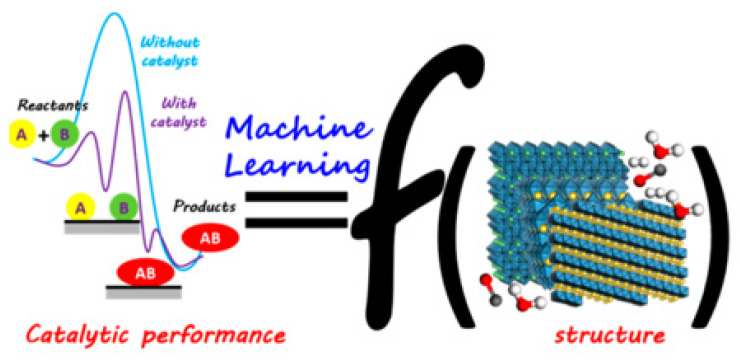
The application of machine learning in material screening. Reprinted with permission from Ref. [235]. Copyright 2020, copyright the American Chemical Society. More details on Copyright and Liscensing are available via the following link: https://pubs.acs.org/.

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
