# Peer review of "A Review of Oil–Solid Separation and Oil–Water Separation in Unconventional Heavy Oil Production Process"

_ijms, 2022, doi:10.3390/ijms24010074_

Round 1
Reviewer 1 Report
Unconventional heavy oil (UHO) is an important petroleum resource. Both oil-solid separation (OSS) and oil-water separation (OWS) processes are key the industrial separation process of UHO. In this paper, an overview of the fundamental theories and research status of oil-solid separation and oil-water separation mechanisms are reviewed and also share our views on the challenges and perspectives for the research of oil-solid separation and oil-water separation in UHO field. The content of this paper is interesting for researchers. However, the paper lacks good organization and cannot be considered for publication in the present form. The points listed below are raised from the reviewer as queries and concerns.
1. The article is too long, so the introduction should be concise and there is no need to introduce some commonsense content. It is unnecessary to quote the experimental results, such as Fig. 13, Fig. 16, Fig. 22, Fig. 37, etc. There are repeated references, for example, reference 26 appears on p9 and p30 respectively.
2. The abstract should be further condensed and rewritten, please check P2,line 26, “Additionally, be directed against the gaps or challenges for OSS and OWS processes. we put forward corresponding resolution strategies from the aspects of the construction of oil-solid interaction mechanism system, the construction of oil-water interaction mechanism system, the construction of separation mechanism system, and the research and development of novel extractants and demulsifiers.”
3. Some expressions are worth discussing, please check carefully such as p. 16, line 324 “Although emulsifying the oil enhances oil separation from UHO, these emulsions are often detrimental to downstream oil-water separation, even leading to overall low oil recovery and poor oil quality [68].”
4. p18, line 368, “The modification of the “Yen-Mullins” is considered one of the most acceptable molecular structures of asphaltenes (Figure 9) [92].” Yen-Mullins model (p.18, Fig. 9) should be molecular and colloidal structures of asphaltenes.
5. P25, line 476, “Because asphaltene belongs to autopolymer, the methods and solvents used to separate UHO will affect the molecular weight of asphaltene, resulting in different interface properties of asphaltene interfacial film.” Is there such a causal relationship?
Author Response
Comments:
Unconventional heavy oil (UHO) is an important petroleum resource. Both oil-solid separation (OSS) and oil-water separation (OWS) processes are key the industrial separation process of UHO. In this paper, an overview of the fundamental theories and research status of oil-solid separation and oil-water separation mechanisms are reviewed and also share our views on the challenges and perspectives for the research of oil-solid separation and oil-water separation in UHO field. The content of this paper is interesting for researchers. However, the paper lacks good organization and cannot be considered for publication in the present form. The points listed below are raised from the reviewer as queries and concerns.
1) The article is too long, so the introduction should be concise and there is no need to introduce some commonsense content. It is unnecessary to quote the experimental results, such as Fig. 13, Fig. 16, Fig. 22, Fig. 37, etc. There are repeated references, for example, reference 26 appears on p9 and p30 respectively.
Many thanks for the reviewer’s comment. Some commonsense content have been revised in the introduction (Section 1, pages 3~8 in the revised manuscript). In addition, all figures and references of the paper have also been rearranged.
2) The abstract should be further condensed and rewritten, please check P2,line 26, “Additionally, be directed against the gaps or challenges for OSS and OWS processes. we put forward corresponding resolution strategies from the aspects of the construction of oil-solid interaction mechanism system, the construction of oil-water interaction mechanism system, the construction of separation mechanism system, and the research and development of novel extractants and demulsifiers.”
We thank the reviewer for this good question. The abstract has been further condensed and rewritten. The specific modifications are as follows. (Page 2 in the revised manuscript)
Unconventional heavy oil ores (UHO) have been considered to be an important part of petroleum resources and alternative source of chemicals and energy supply. Due to the participation of water and extractants, oil-solid separation (OSS) and oil-water separation (OWS) processes are inevitable in the industrial separation process of UHO. Therefore, this critical review systematically reviews basic theories of OSS and OWS, including solid wettability, contact angle, oil-solid interactions, structural characteristics of natural surfactants and interface characteristics of interfacially active asphaltene film. According to the basic theories, the corresponding OSS and OWS mechanisms have been discussed. Finally, the present challenges and future research considerations have been touched to provide insights and theoretical fundamentals for OSS and OWS . Additionally, this critical review would be even useful to provide a framework of research prospects which would guide the future research direction in laboratories and industries focusing on the OSS and OWS processes in this important heavy oil production field.
3) Some expressions are worth discussing, please check carefully such as p. 16, line 324 “Although emulsifying the oil enhances oil separation from UHO, these emulsions are often detrimental to downstream oil-water separation, even leading to overall low oil recovery and poor oil quality [68].”
Thanks the reviewer for this excellent suggestion. We have checked expressions and sentence structure carefully. We believe the revised MS would be much more readable and accurate.
4) p18, line 368, “The modification of the “Yen-Mullins” is considered one of the most acceptable molecular structures of asphaltenes (Figure 9) [92].” Yen-Mullins model (p.18, Fig. 9) should be molecular and colloidal structures of asphaltenes.
Thanks the reviewer again for bringing this good question. The corresponding contents have been revised again. (Page 16 in the revised manuscript)
5) P25, line 476, “Because asphaltene belongs to autopolymer, the methods and solvents used to separate UHO will affect the molecular weight of asphaltene, resulting in different interface properties of asphaltene interfacial film.” Is there such a causal relationship?
Good comments from the reviewer. After another careful checking, we have corrected this sentence as: “Asphaltene belongs to autopolymer, the methods and solvents used to separate UHO will affect the molecular weight of asphaltene, resulting in different interface properties of asphaltene interfacial film”. (Page 23 in revised manuscript)

Reviewer 2 Report
Recommendation: Publish after minor revisions noted.
This work concentrates on the review of oil-solid separation and oil-water separation in unconventional heavy oil production process, which is of great significance to oil exploitation. This review systematically reviews basic theories of OSS and OWS, including solid wettability, contact angle, oil-solid interactions, structural characteristics of natural surfactants and interface characteristics of interfacially active asphaltene film. After going through your paper, I have some comments for you in order to improve your work.
1. The abstract should be concise and state the highlights of the review.
2. line 26 to line 27: “, be directed against the gaps or challenges for OSS and OWS processes. we put forward corresponding resolution strategies……” the Punctuation is used incorrectly.
3. Generally speaking, the “review” should not be the key words, it is suggested to change to other keywords.
4. Please add a space between the value and its unit. Check throughout the draft.
5. Some of the fonts in the images are small and difficult to read. The author should check the full draft images to make changes.
6. The grammar should be carefully revised before the manuscript is published.
7. They have done a lot of research on oil-solid separation for example contact angle alteration, and authors should refer to their research results. “Application of nanomaterial for enhanced oil recovery, petroleum science, https://doi.org/10.1016/j.petsci.2021.11.011”, “Mechanism study of spontaneous imbibition with lower-phase nano-emulsion in tight reservoirs, journal of petroleum science and engineering, https://doi.org/10.1016/j.petrol.2022.110220” and “Laboratory study and field application of amphiphilic molybdenum disulfide nanosheets for enhanced oil recovery, journal of petroleum science and engineering, https://doi.org/10.1016/j.petrol.2021.109695”

Author Response
Comments:
Recommendation: Publish after minor revisions noted.
This work concentrates on the review of oil-solid separation and oil-water separation in unconventional heavy oil production process, which is of great significance to oil exploitation. This review systematically reviews basic theories of OSS and OWS, including solid wettability, contact angle, oil-solid interactions, structural characteristics of natural surfactants and interface characteristics of interfacially active asphaltene film. After going through your paper, I have some comments for you in order to improve your work.
1) The abstract should be concise and state the highlights of the review.
We thank the reviewer for this good question. The abstract has been further condensed and rewritten. The specific modifications are as follows. (Page 2 in the revised manuscript)
Unconventional heavy oil ores (UHO) have been considered to be an important part of petroleum resources and alternative source of chemicals and energy supply. Due to the participation of water and extractants, oil-solid separation (OSS) and oil-water separation (OWS) processes are inevitable in the industrial separation process of UHO. Therefore, this critical review systematically reviews basic theories of OSS and OWS, including solid wettability, contact angle, oil-solid interactions, structural characteristics of natural surfactants and interface characteristics of interfacially active asphaltene film. According to the basic theories, the corresponding OSS and OWS mechanisms have been discussed. Finally, the present challenges and future research considerations have been touched to provide insights and theoretical fundamentals for OSS and OWS . Additionally, this critical review would be even useful to provide a framework of research prospects which would guide the future research direction in laboratories and industries focusing on the OSS and OWS processes in this important heavy oil production field.
2) line 26 to line 27: “, be directed against the gaps or challenges for OSS and OWS processes. we put forward corresponding resolution strategies……” the Punctuation is used incorrectly.
Thanks the reviewer again for bringing this good question. The corresponding contents have been revised again. (Page 2 in the revised manuscript)
3) Generally speaking, the “review” should not be the key words, it is suggested to change to other keywords.
Good comments from the reviewer. We have revised the keywords. The specific modifications are as follows. (Page 2 in revised manuscript)
Keywords: Oil-solid separation; Oil-water separation; Heavy oil ores; Production process
4) Please add a space between the value and its unit. Check throughout the draft.
Thanks the reviewer for this excellent suggestion. After another careful checking, we have added a space between the value and its unit. The corresponding contents have been revised in the manuscript.
5) Some of the fonts in the images are small and difficult to read. The author should check the full draft images to make changes.
Thanks the reviewer. We have checked the full draft images and maked changes in revised manuscript.
6) The grammar should be carefully revised before the manuscript is published.
Thanks the reviewer for this excellent suggestion again. After another careful checking the grammar. We have tried our best to polish the language again by removing some redundant content. We believe the revised MS would be much more readable and accurate.
7) They have done a lot of research on oil-solid separation for example contact angle alteration, and authors should refer to their research results. “Application of nanomaterial for enhanced oil recovery, petroleum science, https://doi.org/10.1016/j.petsci.2021.11.011”, “Mechanism study of spontaneous imbibition with lower-phase nano-emulsion in tight reservoirs, journal of petroleum science and engineering, https://doi.org/10.1016/j.petrol.2022.110220” and “Laboratory study and field application of amphiphilic molybdenum disulfide nanosheets for enhanced oil recovery, journal of petroleum science and engineering, https://doi.org/10.1016/j.petrol.2021.109695”
Good comments from the reviewer. We have cited these three valuable research papers (https://doi.org/10.1016/j.petsci.2021.11.011;https://doi.org/10.1016/j.petrol.2022.110220; https://doi.org/10.1016/j.petrol.2021.109695 ) in revised manuscript. (Page 9 in revised manuscript)

Reviewer 3 Report
The article does not fall within the thematic scope of the journal International Journal of Molecular Sciences. The journal is an peer-reviewed providing an advanced forum for biochemistry, molecular and cellular biology, molecular biophysics, molecular medicine and all aspects of molecular research in chemistry.
The article should be published in its current form in another journal.
The article does not contain factual errors.
Author Response
Comments:
The article does not fall within the thematic scope of the journal International Journal of Molecular Sciences. The journal is an peer-reviewed providing an advanced forum for biochemistry, molecular and cellular biology, molecular biophysics, molecular medicine and all aspects of molecular research in chemistry.
The article should be published in its current form in another journal.
The article does not contain factual errors.
Thanks the reviewer. Although the topic of the article belongs to the separation of heavy oil ores, the content of the article also involves some knowledge of molecular chemistry, such as the interaction of organic chemical molecules in heavy oil. Therefore, the article does fall within the thematic scope of the journal International Journal of Molecular Sciences.

Round 2
Reviewer 1 Report
The author made a good revision and explanation of the questions raised by the reviewer, which is worthy of publishing.
Reviewer 3 Report
The described partial interactions are not the main subject of the article, therefore the review of oil-solid and oil-water separation in the unconventional process of heavy oil production for molecular chemistry cannot be included. The article should be published in another journal.